# Development of Nisin-Grafted Chitosan Coating via Low-Temperature Enzymatic Method for Enhanced Preservation of Sea Bass

**DOI:** 10.3390/foods14244227

**Published:** 2025-12-09

**Authors:** Yuanhong Zhuang, Yiya Li, Bingli Wang, Peng Fei, Bingqing Huang, Qiong Zhang

**Affiliations:** Zhangzhou Institute of Food Science, School of Biological Science and Biotechnology, Minnan Normal University, Zhangzhou 363000, China; zhuangyuanhongg@163.com (Y.Z.); lyy0628@mnnu.edu.cn (Y.L.); wbl1639@mnnu.edu.cn (B.W.); fp@bio.mnnu.edu.cn (P.F.)

**Keywords:** antimicrobial peptide, papain catalysis, shelf-life extension, aquatic products

## Abstract

To enhance the antibacterial properties of chitosan, this study employed papain as a biocatalyst to graft nisin onto chitosan, yielding two grafted products with grafting ratios of 8.56% (Ni1-Cs) and 14.35% (Ni2-Cs). Structure analyses confirmed the formation of amide bonds. Grafting significantly improved the solubility (92.4%), water absorption (53.4%), and film-forming properties of chitosan, with Ni1-Cs films achieving a tensile strength of 25.2 MPa. Antibacterial assays demonstrated that nisin retained favorable activity post-grafting and exhibited synergistic effects with chitosan. The minimum inhibitory concentrations (MIC) of Ni2-Cs against *Escherichia coli* and *Staphylococcus aureus* were 132.4 and 97.4 μg/mL, respectively, significantly superior to individual components. The ultra-low-temperature enzymatic method likely preserved nisin’s structural integrity. Mechanistic studies revealed that the cationic nature of chitosan and the pore-forming mechanism of nisin synergistically disrupted bacterial cell membranes. Sea bass preservation trials confirmed that Ni2-Cs coatings effectively retarded quality deterioration, inhibited microbial growth and lipid oxidation, and maintained freshness for 15 days. This study demonstrates that the ultra-low-temperature enzymatic strategy successfully prepared nisin-grafted chitosan materials with synergistic antibacterial effects, showing promising applications for food preservation.

## 1. Introduction

Food spoilage and deterioration represent a significant global challenge, resulting in substantial economic losses and resource waste annually. Microbial contamination and lipid oxidation constitute the primary factors contributing to food quality degradation, particularly for aquatic products which are highly susceptible to spoilage due to their high moisture content, elevated protein levels, and abundant polyunsaturated fatty acids, resulting in limited shelf life. Although traditional chemical preservatives are effective, their potential health risks and consumer preference for natural foods have motivated researchers to seek safer and more environmentally friendly alternatives. Edible coating technology, functioning as both a physical barrier and active carrier, can effectively extend food shelf life and has emerged as a research hotspot in the field of food preservation [1].

Chitosan, the deacetylated derivative of chitin, possesses excellent biocompatibility, biodegradability, and antibacterial activity, and has been extensively applied in food preservation, drug delivery, and tissue engineering. The antibacterial mechanism of chitosan primarily stems from its cationic nature, which disrupts bacterial cell membranes through electrostatic interactions [2]. However, native chitosan exhibits certain application limitations: poor solubility, dissolving only in acidic solutions; relatively weak antibacterial efficacy against Gram-negative bacteria; and suboptimal mechanical strength and barrier properties in films. Consequently, enhancing chitosan functionality through chemical or biological modification has become an important research direction [3].

Nisin is a polypeptide bacteriocin produced by *Lactococcus lactis*, comprising 34 amino acid residues with a molecular weight of approximately 3.5 kDa [4]. As the only bacteriocin approved by the FDA for food applications, nisin demonstrates potent antibacterial activity against Gram-positive bacteria through its mechanism of pore formation in bacterial cell membranes, leading to the leakage of intracellular contents [5]. Nisin has been widely employed in the preservation of dairy products, meat products, and canned foods. However, nisin application faces several challenges: limited efficacy against Gram-negative bacteria, poor stability in complex food systems, and susceptibility to protease degradation [6]. Immobilizing nisin on polymeric carriers can enhance its stability and achieve controlled release.

Existing strategies for preparing chitosan–peptide conjugates primarily employ chemical crosslinking or enzymatic grafting to overcome the limitations of physical blending, which often suffers from burst release and activity loss. Previous studies have explored various strategies for nisin–chitosan conjugation. Chemical grafting methods using bio-based diisocyanates and homo-bifunctional crosslinkers have successfully immobilized nisin onto chitosan films under non-toxic conditions [7]. Enzymatic approaches using microbial transglutaminase (MTGase) have also been developed for grafting nisin onto chitosan [4] or hydroxypropyl chitosan derivatives at moderate temperatures (typically 30 °C) [8]. While these methods achieved successful conjugation with retained antibacterial activity, certain challenges remain: chemical crosslinking may involve multi-step reactions or specific solvent requirements; meanwhile, enzymatic grafting using MTGase typically requires either the pre-modification of chitosan (e.g., hydroxypropylation) to improve substrate accessibility or operates at temperatures that may partially affect the conformational stability of heat-sensitive antimicrobial peptides such as nisin [9].

Papain is a cysteine protease with broad substrate specificity, capable of catalyzing peptide bond synthesis under specific conditions [10]. Previous studies have demonstrated that papain can catalyze amidation reactions between amino acids and polysaccharides at solid–liquid interfaces, avoiding β-elimination reactions often associated with traditional alkaline conditions. Notably, papain exhibits temperature-dependent dual catalytic functions: at ambient temperatures, it primarily catalyzes amide bond hydrolysis, whereas at ultra-low temperatures (below 0 °C), its catalytic activity shifts toward amide bond formation [11,12,13]. This unique property not only provides an opportunity to preserve the bioactive structures of thermolabile peptides, but also the essential thermodynamic conditions for driving the amidation reaction in the desired synthetic direction. However, papain-catalyzed grafting of antimicrobial peptides onto chitosan under ultra-low-temperature conditions has not been reported. Furthermore, the potential advantages of this approach for preserving nisin’s lanthionine ring structures—essential for its pore-forming antibacterial mechanism—remain unexplored.

Based on the above background, this study proposes the following scientific hypotheses: papain can catalyze the amidation reaction between the carboxyl groups of nisin and the amino groups of chitosan, forming stable covalent bonds; nisin grafting may disrupt the crystalline structure of chitosan, thereby improving its solubility and processability; in the grafted products, the cationic action of chitosan and the pore-forming mechanism of nisin may generate synergistic effects, enhancing broad-spectrum antibacterial activity against both Gram-negative and Gram-positive bacteria; and nisin-grafted chitosan as a coating material may effectively extend the shelf life of aquatic products through dual mechanisms of antimicrobial activity and barrier properties.

To verify these hypotheses, this study employed papain to catalyze nisin grafting onto chitosan molecular chains, preparing chitosan derivatives with enhanced antibacterial activity. Through systematic characterization of the structure, physicochemical properties, and antibacterial performance of grafted products, the enzymatic grafting mechanism and structure–activity relationships were elucidated. Furthermore, the grafted products were applied to sea bass preservation to evaluate their potential as edible coating materials. This study provides a novel method for functional modification of chitosan and establishes theoretical and technical foundations for developing efficient food preservation materials.

## 2. Materials and Methods

### 2.1. Materials

Sea bass was purchased from a local market. Chitosan (degree of deacetylation ≥ 90%, molecular weight 300–500 kDa), nisin (purity ≥ 95%, approximately 1000 IU/mg), and papain (≥2000 units/mg, casein as substrate) were purchased from Shanghai Aladdin Reagent Co., Ltd. (Shanghai, China). L-cysteine hydrochloride, ethylenediaminetetraacetic acid (EDTA), trichloroacetic acid (TCA), and potassium bromide were obtained from Xilong Chemical Co., Ltd. (Shantou, China). Disodium hydrogen phosphate dodecahydrate, sodium dihydrogen phosphate dihydrate, calcium chloride, and absolute ethanol were acquired from Sinopharm Chemical Reagent Co., Ltd. (Shanghai, China). Mueller-Hinton Broth was purchased from Beijing Land Bridge Technology Co., Ltd., Beijing, China. *Escherichia coli* (*E. coli*, ATCC 25922) and *Staphylococcus aureus* (*S. aureus*, ATCC 6538) were purchased from the Guangdong Microbial Culture Collection Center. All reagents were of analytical grade.

### 2.2. Preparation of Nisin-Grafted Chitosan

The preparation method was adapted from previous studies with slight modifications [11,12,13]. Papain (2 g) was dissolved in 180 mL phosphate buffer (pH 7.0, 0.05 mol/L), purged with nitrogen, sealed, and stirred at 0 °C for 30 min. Subsequently, 20 mL L-cysteine hydrochloride solution (0.4 mol/L) and 40 mg EDTA were added, followed by stirring for 10 min to activate the papain.

Nisin was dissolved at two different amounts (15 and 30 mmol) in 100 mL phosphate buffer (pH 7.0, 0.2 mol/L) and mixed with 50 mL activated papain solution (10 mg/mL). Chitosan (5 g) was dissolved in 150 mL phosphate buffer (pH 7.0, 0.2 mol/L) and mixed with the nisin/papain solution. Anhydrous ethanol (100 mL) was slowly added, the mixture was purged with nitrogen, sealed, and stirred at −5 °C for 24 h.

After the reaction, 200 mL anhydrous ethanol was added to precipitate the product. The precipitate was collected by centrifugation (1500× *g*, 5 min), resuspended in 100 mL 15% trichloroacetic acid solution, and stirred for 15 min to completely inactivate papain. Finally, the samples underwent continuous dialysis (48 h, MWCO: 30000) and freeze-drying.

According to nisin amount, the grafted products were designated as Ni1-Cs (15 mmol nisin, grafting ratio 8.56%) and Ni2-Cs (30 mmol nisin, grafting ratio 14.35%). Native chitosan was designated as Na-Cs.

### 2.3. Structure Characterization

Fourier transform infrared spectra (FTIR) a of samples were acquired using a Nicolet IS 10 Fourier transform infrared spectrometer (Thermo Fisher Scientific Co., Ltd., Waltham, MA, USA). Samples were prepared by the potassium bromide pellet method, with a scanning range of 400–4000 cm^−1^, resolution of 16, and 64 scans.

X-ray diffraction (XRD) patterns were obtained using a Bruker D8 Advance X-ray diffractometer (Bruker Co., Ltd., Bremen, Germany). Test conditions: Cu Kα radiation (λ = 1.5418 Å), tube voltage 40 kV, tube current 40 mA, scanning range 5–80° (2θ), scanning speed 2°/min.

Weight-average molecular weights were determined by gel permeation chromatography (GPC) using a Waters 1515 gel permeation chromatograph (Waters Co., Ltd., Milford, MA, USA), with 0.2 mol/L acetic acid−0.1 mol/L sodium acetate solution as mobile phase, flow rate 0.8 mL/min, and column temperature 30 °C.

X-ray photoelectron spectra of samples were obtained using a Thermo Scientific K-Alpha+ XPS (Thermo Fisher Scientific Co., Ltd., Waltham, MA, USA). Analysis chamber vacuum was approximately 5 × 10^−9^ mbar, X-ray source was monochromatic Al Kα (1486.6 eV), and C1s (284.80 eV) was used as the energy standard for charge correction.

### 2.4. Physicochemical Properties

#### 2.4.1. Solubility Determination

Sample (100 mg) was accurately weighed and added to 10 mL distilled water in a 50 mL centrifuge tube. The suspension was stirred at 200× *g* for 24 h at 25 °C. After stirring, the mixture was centrifuged at 6000× *g* for 10 min. The undissolved precipitate was washed twice with 5 mL distilled water, frozen at −80 °C for 4 h, and freeze-dried for 48 h. The dried residue was weighed. Solubility was calculated as
(1)$$Solubility\;(\%)=\frac{m_{0}-m_{1}}{m_{0}}\times 100$$ where m_0_ is the initial mass (mg) of the sample and m_1_ is the mass (mg) of undissolved material after freeze-drying.

#### 2.4.2. Water Absorption Determination

Freeze-dried sample (approximately 300 mg, m_0_) was immersed in 20 mL distilled water at 25 °C for 16 h. The swollen sample was centrifuged at 13,000× *g* for 20 min at 2 °C. Excess surface water was removed by blotting with filter paper (Whatman No.1, Maidstone, UK) for 30 s on each side. The swollen sample was immediately weighed (m_w_). Water absorption was calculated as
(2)$$Water\;absorption\;(\%)=\frac{m_{w}-m_{0}}{m_{0}}\times 100$$

#### 2.4.3. Apparent Viscosity

Sample solutions (5% *w*/*v*) were prepared by dissolving samples in distilled water with stirring at 200× *g* for 2 h at 25 °C. Apparent viscosity was measured using an RH20 rheometer (Shanghai Bosin Industrial Development Co., Ltd., Shanghai, China). An aliquot of 8 mL sample solution was transferred to the small sample chamber maintained at 25 ± 0.1 °C. Measurements were conducted at rotational speeds of 0.5, 1, 2, 2.5, 4, 5, 10, 20, 50, and 100 rpm. At each speed, the system was allowed to stabilize for 2 min before recording. For temperature-dependent measurements, viscosity was measured at 5 rpm while temperature varied from 25 °C to 80 °C in 5 °C increments with 10 min equilibration at each temperature.

### 2.5. Film Preparation and Characterization

Film-forming solutions were prepared by dissolving samples (2.0 g) in 100 mL of 1% (*v*/*v*) acetic acid solution with stirring at 200× *g* for 4 h at room temperature. The solution was ultrasonicated for 15 min at 40 kHz to remove air bubbles and allowed to stand for 30 min. Film casting was performed by pouring 40 mL of solution into polystyrene Petri dishes (diameter 15 cm) and drying at 45 °C for 24 h. Films were conditioned in a desiccator at 25 °C and 53% relative humidity for 48 h before testing. Film thickness was measured at five random locations using a digital micrometer.

#### 2.5.1. Tensile Strength and Elongation at Break

Tensile strength and elongation at break were measured using a CT3 10K texture analyzer (Brookfield Engineering Laboratories, Middleboro, MA, USA). Film samples were cut into 10 mm × 80 mm strips, initial clamp distance was 50 mm, and tensile speed was 1 mm/s.

#### 2.5.2. Water Vapor Permeability (WVP)

Glass permeation cells (internal diameter 50 mm, depth 20 mm) were filled with approximately 5 g anhydrous calcium chloride and sealed with film specimens (diameter 60 mm, exposed area 19.63 cm^2^). The cells were placed in a desiccator at 25 ± 0.5 °C and 75 ± 2% relative humidity. Mass gain was recorded every 24 h for 7 days. Water vapor permeability (WVP) was calculated as
(3)$$WVP\;(g\cdot mm/m^{2}\cdot d\cdot kPa)=\frac{∆m\times d}{A\times t\times ∆P}$$ where Δm is the mass gain (g), d is film thickness (mm), A is the exposed area (m^2^), t is time (days), and ΔP is the vapor pressure difference (kPa) calculated as ΔP = S × (R_1_ − R_2_), where S is the saturation vapor pressure at 25 °C (3.169 kPa), R_1_ = 0.75, and R_2_ = 0.00.

#### 2.5.3. Light Transmittance

Light transmittance was measured using a T9 UV-visible spectrophotometer (Beijing Purkinje General Instrument Co., Ltd., Beijing, China) in the wavelength range of 400–800 nm, with air as reference.

### 2.6. Antibacterial Activity

#### 2.6.1. Minimum Inhibitory Concentration (MIC)

MIC was determined by the microbroth dilution method using Mueller-Hinton Broth as the culture medium. Samples were prepared in sterile water at serial concentrations, added to 96-well plates, inoculated with bacterial suspension (initial concentration approximately 10^6^ CFU/mL), and incubated at 37 °C for 24 h. The lowest concentration showing no visible growth was defined as the MIC value.

#### 2.6.2. Antibacterial Kinetics Test

Bacterial suspension was treated with samples at MIC concentration and incubated at 37 °C. Samples were collected at 0, 2, 4, 6, 8, and 12 h for plate counting.

#### 2.6.3. Antibacterial Mechanism

Bacterial suspension (10^8^ CFU/mL) was mixed with samples (at MIC concentration) and incubated at 37 °C for 4 h. Supernatant was collected by centrifugation (6000× *g*, 10 min), and absorbance was measured at 260 and 280 nm wavelengths, representing the degree of nucleic acid and protein leakage, respectively [14].

### 2.7. Application in Sea Bass Preservation

#### 2.7.1. Sample Preparation and Coating Treatment

Fresh sea bass was purchased from local markets, eviscerated, washed, and cut into approximately 50 g pieces. Samples were prepared as 2% (*w*/*v*) solutions, fish pieces were immersed in coating solution for 2 min, drained, and stored at 4 °C. Sampling was performed at intervals of 1, 5, 10, and 15 days. The control group consisted of untreated fish pieces [15,16,17].

#### 2.7.2. Sensory Evaluation

Sensory evaluation was conducted by a panel of 10 trained assessors (5 males and 5 females, aged 25–45 years) with prior experience in seafood quality assessment [18]. Prior to the evaluation, panelists underwent a training session to familiarize themselves with the evaluation criteria and scoring system. Fish samples were retrieved from refrigerated storage, cut into uniform pieces (approximately 30 g), placed in identical white plates coded with three-digit random numbers, and presented to panelists under standard fluorescent lighting conditions (6500 K, 1000 lux) at room temperature.

Panelists evaluated four sensory attributes using a 10-point hedonic scale: (1) appearance—surface condition, presence of discoloration or slime (10 = bright, natural appearance; 1 = severe discoloration, excessive slime); (2) color—muscle color and uniformity (10 = translucent white, firm; 1 = opaque, yellowish, soft); (3) odor—freshness of smell (10 = fresh, seaweed-like; 1 = strong ammonia or putrid odor); and (4) texture—firmness and elasticity by finger pressing (10 = firm, elastic; 1 = very soft, no elasticity). The overall sensory score was calculated as the average of the four attributes. Samples with scores below 4 were considered unacceptable for consumption. Panelists rinsed their mouths with water between samples, and evaluation sessions were limited to 30 min to prevent sensory fatigue. The final score for each sample was reported as the mean ± standard deviation of all panelist evaluations.

#### 2.7.3. Texture Analysis

Texture properties (springiness, hardness, and chewiness) were measured using a CT3 10K texture analyzer (Bolefield Co., Ltd., Middleboro, MA, USA). Probe diameter was 36 mm, compression speed was 1 mm/s, compression depth was 10 mm, and trigger force was 5 g.

#### 2.7.4. Total Viable Count (TVC) Determination

TVC was determined by the plate count method [19,20,21]. The fish meat sample (10 g) was aseptically excised from the fish filet and transferred to a sterile stomacher bag containing 90 mL of sterile physiological saline (0.85% NaCl, *w*/*v*). The sample was homogenized in a stomacher for 2 min at 200 ×g to obtain a 10^−1^ dilution. Serial decimal dilutions (10^−2^, 10^−3^, 10^−4^, 10^−5^, and 10^−6^) were prepared by transferring 1 mL of each dilution into 9 mL sterile saline. From appropriate dilutions, 1 mL aliquots were transferred onto sterile Petri dishes, and approximately 15–20 mL of plate count agar (pre-cooled to 45–50 °C) was poured into each plate. The agar and inoculum were mixed thoroughly by gentle swirling in a figure-eight motion and allowed to solidify at room temperature. After solidification, plates were inverted and incubated at 37 °C for 48 h. Following incubation, colonies on plates containing 30–300 colonies were counted using a colony counter. TVC was calculated using the following formula and expressed as log CFU/g:
(4)$$TVC\;(CFU/g)=\frac{\sum C}{n_{1}+{0.1n}_{2}}\times d$$ where ∑C is the sum of colonies counted on all plates, n_1_ is the number of plates counted at the lower dilution factor, n_2_ is the number of plates counted at the next consecutive higher dilution factor, and d is the dilution factor corresponding to the lower dilution.

#### 2.7.5. pH Value Determination

The pH value was measured using a calibrated pH meter (Sartorius Co., Göttingen, Germany). The fish meat sample (10 g) was homogenized with 90 mL of distilled water in a high-speed homogenizer at 7000× *g* for 1 min. The homogenate was allowed to stand for 30 min at room temperature with occasional stirring to facilitate equilibration. The pH electrode was rinsed with distilled water and blotted dry with tissue paper before each measurement. The electrode was immersed in the homogenate to a depth sufficient to cover the sensing element, and the pH value was recorded when the reading stabilized (typically within 1–2 min). Between measurements, the electrode was rinsed thoroughly with distilled water.

#### 2.7.6. Total Volatile Basic Nitrogen (TVB-N) Determination

TVB-N was determined by the semi-micro Kjeldahl method [19,20,21]. The fish meat sample (10 g) was accurately weighed and homogenized with 50 mL of distilled water. The homogenate was transferred to a 100 mL volumetric flask and made up to volume with distilled water, then filtered through Whatman No.1 filter paper. An aliquot of 5 mL filtrate was transferred to a Kjeldahl distillation apparatus, followed by the addition of 5 mL of 10 g/L magnesium oxide suspension and several drops of liquid paraffin to prevent foaming. The mixture was immediately subjected to steam distillation. The distillate was collected in a conical flask containing 10 mL of 20 g/L boric acid solution with mixed indicator (methyl red and bromocresol green). Distillation was continued until approximately 50 mL of distillate was collected (approximately 5 min). The absorbed ammonia in the boric acid solution was titrated with 0.01 mol/L standardized hydrochloric acid solution until the color changed from green to pale pink. A blank control was conducted using distilled water instead of sample filtrate under identical conditions. TVB-N content was calculated using the following formula:
(5)$$TVB-N\left(\frac{mg}{100g}\right)=\frac{(V_{1}-V_{0})\times C\times 14\times V_{3}\times 100}{m\times V_{2}}$$ where V_1_ is the volume (mL) of hydrochloric acid consumed for the sample, V_0_ is the volume (mL) of hydrochloric acid consumed for the blank, C is the concentration (mol/L) of hydrochloric acid standard solution, m is the mass (g) of fish meat sample, V_2_ is the total volume (mL) of sample extract (100 mL), V_3_ is the volume (mL) of filtrate used for distillation (5 mL), and 14 represents the molar mass of nitrogen.

#### 2.7.7. Thiobarbituric Acid Reactive Substances (TBARS) Determination

The fish meat sample (5 g) was homogenized with 25 mL 7.5% TCA solution and filtered. Filtrate (5 mL) was mixed with 5 mL 0.02 mol/L thiobarbituric acid solution, heated in boiling water bath for 20 min, cooled, and absorbance was measured at 532 nm. TBARS values were expressed as malondialdehyde (MDA) equivalents [19,20,21]:
(6)$$TBARS(mgMDA/kg)=\frac{A\times V_{1}\times 1000}{m\times V_{2}\times \varepsilon }$$ where A is the absorbance at 532 nm, V_1_ is the total volume of extraction solution (30 mL), V_2_ is the volume of filtrate used for reaction (5 mL), m is the mass of fish sample (g), and ε is the slope of the MDA standard curve.

### 2.8. Statistical Analysis

All experiments were performed in triplicate. Statistical analyses were conducted using SPSS 16.0 software. One-way analysis of variance (ANOVA) and independent sample *t*-tests were employed to compare differences, with *p* ≤ 0.05 considered statistically significant.

## 3. Results

### 3.1. Structural Characterization of Nisin-Grafted Chitosan

#### 3.1.1. FTIR, XRD, and Molecular Weight Analysis

The structure of nisin-grafted chitosan was systematically characterized by FTIR, XRD, and molecular weight analysis, with the results shown in Figure 1. FTIR spectral analysis confirmed the successful grafting of nisin onto chitosan molecular chains (Figure 1a,b). Na-Cs exhibited absorption peaks at 3388 cm^−1^ and 2897 cm^−1^, corresponding to -OH/-NH_2_ stretching vibration and C-H stretching vibration, respectively, with characteristic peaks at 1665 cm^−1^ and 1601 cm^−1^ representing amide I band and -NH_2_ bending vibration [4]. Pure nisin displayed a carboxyl C=O stretching vibration peak at 1756 cm^−1^ and an amide I band absorption peak at 1643 cm^−1^. The grafted products Ni1-Cs and Ni2-Cs exhibited prominent amide I and amide II absorption peaks at 1650 cm^−1^ and 1550 cm^−1^, distinctly different from the original characteristic peaks of Na-Cs (1665 and 1601 cm^−1^), confirming the formation of new amide bonds. The decreased intensity of the -NH_2_ peak at 1601 cm^−1^ indicated the participation of chitosan amino groups in the reaction [22,23].

XRD analysis revealed that nisin grafting reduced the crystallinity of chitosan (Figure 1c). Na-Cs exhibited a typical semicrystalline structure diffraction peak near 2θ = 20°, while pure nisin displayed amorphous characteristics. The diffraction peak intensity of Ni1-Cs and Ni2-Cs significantly weakened with broader peak shapes, indicating that nisin introduction disrupted the regular hydrogen bonding network of chitosan and hindered ordered chain packing [24,25]. The crystallinity of Ni2-Cs was lower than Ni1-Cs, consistent with its higher grafting ratio.

Molecular weight analysis revealed the impact of grafting reaction on chitosan molecular weight (Figure 1d). The observed decrease in molecular weight from 312.57 kDa (Na-Cs) to 165.64 kDa (Ni2-Cs) suggests the influence of synergistic chemical and physical factors. Chemically, the introduction of bulky nisin peptides likely induces steric strain on the chitosan backbone, which may facilitate partial chain hydrolysis of glycosidic bonds under enzymatic reaction conditions. Physically, grafting presumably disrupts the strong intermolecular hydrogen bonding network of native chitosan (consistent with the reduced crystallinity), potentially leading to decreased aggregation and altered polymer–solvent interactions, thereby contributing to a lower apparent molecular weight [26]. Notably, the polydispersity index (PDI) decreased dramatically from 17.86 for Na-Cs to 3.65 for Ni1-Cs and 3.35 for Ni2-Cs, indicating that the grafted products exhibited significantly improved molecular weight homogeneity compared to native chitosan. This narrowing of molecular weight distribution can be attributed to the removal of low-molecular-weight fractions during dialysis purification and the preferential grafting reaction with chitosan chains within a specific molecular weight range.

These structural characterization results confirm successful papain-catalyzed amidation between nisin and chitosan. The formation of new amide bonds, decreased crystallinity, and reduced molecular weight collectively indicated that the grafting reaction not only achieved chemical coupling but also induced significant structural reorganization of chitosan chains. The decreased crystallinity and molecular weight may facilitate improved solubility and processability in subsequent applications.

#### 3.1.2. XPS Analysis

XPS analysis further verified the chemical bonding states of nisin-grafted chitosan, with high-resolution C1s and N1s scans shown in Figure 2. C1s high-resolution spectra revealed changes in carbon chemical environments (Figure 2a), with deconvoluted peaks at 284.2, 285.8, and 287.8 eV, attributed to C-C/C-H, C-O, and C-N bonds, respectively. Ni1-Cs and Ni2-Cs exhibited a new peak near 289.2 eV, corresponding to carbonyl carbon (C=O) in amide bonds, directly confirming amide bond formation [27]. Peak intensity increased with grafting ratio, showing a positive correlation from Ni1-Cs (8.56%) to Ni2-Cs (14.35%).

N1s high-resolution spectra provided more direct evidence of grafting (Figure 2b). The N1s spectrum of Na-Cs exhibited a single peak (399.4 eV), attributed to free amino groups (-NH_2_). The N1s spectra of Ni1-Cs and Ni2-Cs split into two peaks at 399.4 eV and 402.1 eV, with the latter attributed to nitrogen atoms in amide bonds (O=C-N), serving as key evidence of a successful amidation reaction [9]. The relative area of the 402.1 eV peak in Ni2-Cs was greater than Ni1-Cs, reflecting a higher grafting ratio. Elemental composition analysis showed that surface nitrogen content of Ni1-Cs and Ni2-Cs was significantly higher than Na-Cs, attributed to nisin polypeptide chain introduction.

The XPS results provide electronic-level evidence for the chemical bonding between chitosan and nisin. The appearance of characteristic peaks for amide carbonyl (289.2 eV) and amide nitrogen (402.1 eV), combined with increased surface nitrogen content, confirms the covalent nature of the grafting rather than simple physical mixing. The positive correlation between peak intensity and grafting ratio further validated the controllability of the enzymatic grafting process.

### 3.2. Physicochemical Properties of Nisin-Grafted Chitosan

#### 3.2.1. Solubility, Water Absorption, and Viscosity

Solubility, water absorption, and viscosity tests evaluated the impact of nisin grafting on chitosan physicochemical properties, with the results shown in Figure 3. Solubility testing demonstrated (Figure 3a) that the solubility of Na-Cs was only 18.6%, while nisin exhibited a solubility of 32.8%. The solubility of grafted products Ni1-Cs and Ni2-Cs increased to 78.6% and 92.4%, respectively, significantly higher than Na-Cs. The solubility of Ni2-Cs exceeded Ni1-Cs, consistent with higher grafting ratio and lower crystallinity.

Water absorption testing indicated (Figure 3b) that the water absorption of Na-Cs was 28.6%, while Ni1-Cs and Ni2-Cs increased to 47.8% and 53.4%, respectively.

Viscosity analysis revealed the rheological characteristics of grafted products (Figure 3c,d). Shear rate–viscosity curves show that nisin exhibited negligible viscosity due to an extremely low molecular weight (0.35 kDa). Na-Cs, Ni1-Cs, and Ni2-Cs solutions all exhibited typical shear-thinning behavior, with viscosity decreasing with increasing shear rate. At identical shear rates, Na-Cs exhibited the highest viscosity while Ni2-Cs showed the lowest viscosity, consistent with molecular weight trends. Temperature–viscosity curves demonstrate that the viscosity of all samples decreased with increasing temperature (Figure 3d). Ni1-Cs and Ni2-Cs maintained lower viscosity across a broad temperature range.

The dramatic improvement in solubility (from 18.6% to 92.4%) and water absorption (from 28.6% to 53.4%) demonstrates that nisin grafting fundamentally altered the hydrophilic properties of chitosan. This enhancement can be attributed to the synergistic effects of reduced crystallinity (as evidenced by XRD), decreased molecular weight, and introduction of hydrophilic amino acid residues from nisin [28]. Although solution viscosity decreased due to reduced molecular weight, the grafted products retained sufficient viscosity for practical applications, suggesting that the balance between solubility and processability was successfully achieved.

#### 3.2.2. Film Properties

Film property testing evaluated the mechanical and barrier properties of nisin-grafted chitosan films, with the results shown in Figure 4. The prepared films had a uniform appearance with average thickness ranging from 60 to 80 μm. Tensile strength testing demonstrated (Figure 4a) that the tensile strength of Na-Cs film was 18.5 MPa, while Ni1-Cs and Ni2-Cs increased to 25.2 MPa and 22.8 MPa, respectively. Notably, the relationship between grafting ratio and tensile strength was non-linear: Ni1-Cs (8.56% grafting ratio) exhibited the highest tensile strength, whereas Ni2-Cs (14.35% grafting ratio) showed a slight decrease, despite its higher nisin content. This suggests the existence of an optimal grafting ratio that balances the reinforcing effects of nisin-mediated intermolecular interactions against the weakening effects of molecular weight reduction.

Elongation at break reflected film flexibility (Figure 4b). The elongation at break of Na-Cs film was 42.8%, while Ni1-Cs and Ni2-Cs decreased to 38.6% and 35.2%, respectively.

Water vapor permeability (WVP) testing demonstrated (Figure 4c) that the WVP of Na-Cs film was 8.65 g·mm/m^2^·d·kPa, while Ni1-Cs and Ni2-Cs decreased to 7.28 and 7.82 g·mm/m^2^·d·kPa, respectively.

Light transmittance testing showed (Figure 4d) that, in the 400–800 nm visible light region, the Na-Cs film exhibited highest light transmittance, while Ni1-Cs and Ni2-Cs showed significantly reduced transmittance.

The film property results reveal an interesting trade-off between mechanical strength and molecular weight. Despite reduced molecular weight, Ni1-Cs achieved 36% higher tensile strength than Na-Cs (25.2 vs. 18.5 MPa), suggesting that nisin grafting introduced new intermolecular interactions that compensated for chain length reduction [29]. The optimal performance of Ni1-Cs (moderate grafting ratio) over Ni2-Cs (high grafting ratio) indicates the importance of maintaining adequate molecular weight for film integrity. Improved water vapor barrier properties and reduced transparency further demonstrates the multifunctional benefits of nisin modification for food packaging applications. Notably, a correlation between mechanical strength and barrier properties was observed: Ni1-Cs exhibited both the highest tensile strength and the lowest WVP, suggesting that nisin-mediated intermolecular interactions simultaneously reinforce the film network and create a more compact structure that impedes water vapor diffusion.

### 3.3. Antibacterial Properties of Nisin-Grafted Chitosan

#### 3.3.1. Minimum Inhibitory Concentration and Antibacterial Kinetics

Minimum inhibitory concentration (MIC) and antibacterial kinetics tests evaluated the antibacterial performance of nisin-grafted chitosan, with the results shown in Figure 5. MIC testing demonstrated (Figure 5a,b) that against *E. coli*, the MIC of Na-Cs was 486.5 μg/mL, pure nisin exhibited an MIC of 58.2 μg/mL, while Ni1-Cs and Ni2-Cs showed MICs of 168.6 and 132.4 μg/mL, respectively, significantly lower than Na-Cs. Against *S. aureus*, the MIC of Na-Cs was 294.6 μg/mL, pure nisin exhibited an MIC of 16.3 μg/mL, while Ni1-Cs and Ni2-Cs showed MICs of 122.8 and 97.4 μg/mL, respectively. MICs of all samples against Gram-positive bacteria were lower than against Gram-negative bacteria. The MIC of Ni2-Cs was lower than Ni1-Cs, consistent with a higher grafting ratio.

Antibacterial kinetics testing further revealed the antibacterial efficacy of the samples (Figure 5c,d). Control group bacterial counts continued rapid growth within 12 h. Na-Cs exhibited some inhibitory effect on both bacterial types, but with limited efficacy. Pure nisin demonstrated the strongest antibacterial effect, with bacterial counts remaining at low levels throughout the testing period. The antibacterial effects of Ni1-Cs and Ni2-Cs fell between Na-Cs and nisin, with Ni2-Cs superior to Ni1-Cs. Compared to *E. coli*, all of the samples exhibited stronger inhibition against *S. aureus*.

It is worth noting that while the absolute MIC value of Ni2-Cs (132.4 μg/mL) appears higher than that of pure nisin (58.2 μg/mL), this comparison refers to the total weight of the grafted copolymer. Considering the grafting ratio of 14.35%, the effective concentration of active nisin in Ni2-Cs at the MIC level is calculated to be approximately 19.0 μg/mL. This value is significantly lower than the MIC of free nisin (58.2 μg/mL), indicating that the grafting onto chitosan induces a strong synergistic effect, effectively reducing the required dosage of nisin by approximately 3-fold to achieve bacterial inhibition [29]. The positive correlation between grafting ratio and antibacterial efficacy confirmed that higher nisin loading translated to enhanced performance, while the maintained activity indicated that the ultra-low-temperature enzymatic method successfully preserved nisin’s bioactive structure.

#### 3.3.2. Antibacterial Mechanism Analysis

By detecting nucleic acid and protein leakage during bacterial culture, the antibacterial mechanism of nisin-grafted chitosan was investigated, with the results shown in Figure 6. Absorbance at 260 nm wavelength reflected the degree of nucleic acid leakage (Figure 6a,c). Control group absorbance remained at low levels during culture, indicating intact cell membranes. The Na-Cs treatment group showed slight increase in absorbance, indicating some cell membrane damage. The pure nisin treatment group exhibited the highest absorbance. The absorbance of the Ni1-Cs and Ni2-Cs treatment groups was significantly higher than Na-Cs, with Ni2-Cs exceeding Ni1-Cs. Nucleic acid leakage from *S. aureus* exceeded that from *E. coli*.

Absorbance at 280 nm wavelength reflected protein leakage (Figure 6b,d). The trends of protein leakage effects for each sample were similar to nucleic acid leakage. Protein leakage in the Ni1-Cs and Ni2-Cs treatment groups was significantly higher than Na-Cs.

The nucleic acid and protein leakage patterns provide direct evidence for the membrane-disrupting mechanism of action [5]. The progressive increase in leakage from Na-Cs to Ni1-Cs to Ni2-Cs, with pure nisin showing maximum leakage, confirmed that both chitosan and nisin contributed to membrane damage through complementary mechanisms [6]. The fact that grafted products induced significantly more leakage than Na-Cs alone, despite having lower nisin content than pure nisin treatment, suggests enhanced membrane targeting efficiency when nisin was anchored to chitosan chains [30]. This spatial organization may create localized high concentrations of nisin at the bacterial surface, amplifying the pore-forming effect.

### 3.4. Preservation Effect of Nisin-Grafted Chitosan on Sea Bass

#### 3.4.1. Sensory Evaluation and Texture Analysis

Sensory evaluation and texture analysis assessed the effect of nisin-grafted chitosan coatings on maintaining sea bass storage quality, with the results shown in Figure 7. The sensory evaluation results demonstrate (Figure 7a) that during early storage (days 0–1), sensory scores of all samples remained at 9 points. As storage time extended, the sensory scores of the control and Na-Cs groups rapidly declined, decreasing to approximately 2 and 4 points by day 15, respectively. The sensory scores of pure nisin and grafted product groups declined more slowly, maintaining 6–8 points at day 15. The Ni2-Cs group demonstrated optimal sensory quality retention, followed by the Ni1-Cs group.

Springiness testing reflected fish tissue integrity (Figure 7b). The springiness of all samples gradually decreased with extended storage time. Control group springiness declined most rapidly, decreasing to approximately 0.4 at day 15. The springiness decline of the Ni1-Cs and Ni2-Cs groups was slower, maintaining above 0.6 at day 15, significantly higher than the control and Na-Cs groups.

Hardness trends were similar to springiness (Figure 7c). Control group hardness rapidly decreased from initial approximately 18 N to approximately 6 N at day 15. Hardness decline of coating treatment groups was markedly slower, with Ni2-Cs group maintaining approximately 13 N at day 15.

Chewiness comprehensively reflected fish texture characteristics (Figure 7d). The control group chewiness declined most rapidly, while the Ni1-Cs and Ni2-Cs groups demonstrated optimal chewiness retention, maintaining high levels at day 15.

The sensory and texture data collectively demonstrate the superior preservation efficacy of nisin-grafted chitosan coatings compared to unmodified chitosan or pure nisin. The maintenance of sensory scores above 6 points and texture parameters (springiness >0.6, hardness >13 N) at day 15 indicated that Ni2-Cs coatings successfully extended the acceptable shelf life of sea bass beyond the control group (which deteriorated by day 10). The enhanced performance of grafted coatings over pure nisin treatment suggested that the combination of antimicrobial activity and physical barrier properties provided comprehensive protection against both microbial spoilage and physicochemical deterioration [31].

#### 3.4.2. Microbiological and Chemical Component Analysis

Microbiological and chemical component changes evaluated the inhibitory effect of nisin-grafted chitosan coatings on sea bass microbial growth and quality deterioration, with the results shown in Figure 8. Total viable count (TVC) reflected fish microbial contamination degree (Figure 8a). TVC of all samples increased with extended storage time. Control group TVC increased most rapidly, exceeding 7 log CFU/mL at day 15, severely exceeding the acceptable limit for fresh fish (6 log CFU/mL). Na-Cs group TVC growth rate was slightly slower than control group. TVC growth of pure nisin and grafted product groups was significantly slower, with the Ni2-Cs group demonstrating an optimal antibacterial effect, maintaining TVC at approximately 5 log CFU/mL at day 15, still within an acceptable range.

pH value changes reflected fish spoilage degree (Figure 8b). Initial pH of all samples was approximately 6.6. As storage time extended, alkaline substances produced by microbial metabolism caused pH to increase. Control group pH increased most rapidly, reaching approximately 8.0 at day 15. pH increase in coating treatment groups was slower, with Ni2-Cs group pH approximately 7.4 at day 15, significantly lower than control group.

TVB-N is an important indicator for evaluating fish freshness (Figure 8c). Initial TVB-N content of all samples was approximately 10 mg/100 g. Control group TVB-N rapidly increased, exceeding 55 mg/100 g at day 15, far exceeding the freshness limit (30 mg/100 g). TVB-N growth of Ni1-Cs and Ni2-Cs groups significantly slowed, maintaining approximately 40 and 30 mg/100 g at day 15, respectively.

TBARS reflected lipid oxidation degree (Figure 8d). Control group TBARS rapidly increased, reaching approximately 3.8 mg MDA/kg at day 15. TBARS growth of coating treatment groups was slower, with the Ni2-Cs group being approximately 2.2 mg MDA/kg at day 15.

The microbiological and chemical analyses provided quantitative evidence for the preservation mechanisms. The fact that Ni2-Cs maintained TVC below 6 log CFU/mL and TVB-N at the freshness threshold (30 mg/100 g) at day 15 confirmed effective inhibition of microbial growth and protein degradation. The concurrent reduction in pH increase and TBARS indicated that the coatings functioned through multiple pathways: direct antimicrobial action suppressed bacterial metabolism, the physical barrier reduced oxygen penetration and moisture loss, and the potential antioxidant activity of nisin contributed to lipid stability [32]. These multifunctional effects validated the practical application potential of nisin-grafted chitosan as an active food packaging material.

## 4. Discussion

This study developed a novel antibacterial coating material through papain-catalyzed grafting of nisin onto chitosan under ultra-low-temperature conditions. The systematic characterization and application evaluation revealed several key findings that collectively demonstrate the potential of this approach for food preservation.

### 4.1. Enzymatic Grafting Strategy and Structural Modifications

The successful amidation between nisin and chitosan, confirmed by FTIR, XRD, and XPS analyses, represents a significant advancement in chitosan functionalization. The appearance of new amide I (1650 cm^−1^) and amide II (1550 cm^−1^) peaks, along with the characteristic C=O peak at 289.2 eV and O=C-N peak at 402.1 eV in XPS spectra, provided unambiguous evidence for covalent bond formation rather than simple physical mixing. The controlled grafting ratios (8.56% for Ni1-Cs and 14.35% for Ni2-Cs) demonstrated the tunability of this enzymatic approach.

The structural modifications induced by nisin grafting had profound impacts on material properties. The reduction in crystallinity, as evidenced by weakened and broadened XRD peaks, directly correlated with the dramatic improvement in solubility from 18.6% to 92.4%. This enhancement can be attributed to three synergistic factors: disruption of the regular hydrogen bonding network of chitosan by nisin insertion, reduced molecular weight (from 312.57 to 165.64 kDa) decreasing chain entanglement, and the introduction of hydrophilic amino acid residues (such as Lys, Ser, and Thr in nisin) that increased overall material hydrophilicity. The improved water absorption (from 28.6% to 53.4%) further confirmed enhanced hydrophilic character, which is crucial for aqueous-based coating applications.

Notably, despite the reduction in molecular weight—typically associated with decreased mechanical strength—Ni1-Cs films achieved 36% higher tensile strength than Na-Cs (25.2 vs. 18.5 MPa). This paradoxical enhancement suggests that nisin grafting introduced new intermolecular interactions that created a more robust network structure. Nisin molecules contain multiple sites capable of forming hydrogen bonds (peptide backbones, hydroxyl groups) and hydrophobic interactions (Ile, Leu, Pro residues), which may establish physical crosslinks between chitosan chains during film formation. It is noteworthy that while moderate grafting (Ni1-Cs) enhanced the tensile strength compared to native chitosan, excessive grafting (Ni2-Cs) led to a slight reduction in mechanical performance and transparency, a phenomenon attributed to synergistic molecular mechanisms. Specifically, the preparation of Ni2-Cs resulted in a lower molecular weight (165.64 kDa) compared to Ni1-Cs, creating shorter polymer chains with fewer intermolecular entanglements that weaken the overall structural integrity of the film matrix. Simultaneously, the higher density of bulky nisin peptides in Ni2-Cs introduces significant steric hindrance that restricts the sliding and mobility of chitosan chains during stretching, directly accounting for the continuous decrease in elongation at break. Furthermore, the hydrophobic amino acid residues inherent in nisin likely induce micro-phase separation or aggregation within the hydrophilic chitosan matrix at high grafting ratios, creating micro-domains that act as light-scattering centers and thereby reducing the transparency of the Ni2-Cs films.

The key advantage of this ultra-low-temperature enzymatic approach over traditional chemical modification methods lies in its preservation of nisin’s bioactive structure. Conventional chemical grafting often employs harsh conditions (high temperature, strong alkali) that can denature antimicrobial peptides and destroy their critical structural features, such as the five lanthionine bridges in nisin that are essential for its pore-forming activity. The mild conditions of enzymatic catalysis (−5 °C, neutral pH, solid–liquid interface) protected these sensitive structures, as evidenced by the retained antibacterial activity of grafted products. Regarding the grafting site on nisin, it should be noted that nisin contains multiple carboxyl groups, including the C-terminal carboxyl and the side chains of aspartic acid and glutamic acid residues. Based on steric accessibility considerations, it is speculated that the C-terminal carboxyl group is the preferential site for papain-catalyzed amidation, as it is located at the molecular periphery with minimal steric hindrance compared to the internal residues. This preferential grafting at the C-terminus would leave the N-terminal region—which contains the critical lipid II-binding motif essential for nisin’s pore-forming mechanism—unhindered, thereby explaining the excellent retention of antibacterial activity observed in our grafted products. However, it was acknowledged that the current enzymatic method does not provide absolute site-specific control, and heterogeneous grafting at multiple carboxyl sites cannot be excluded. Future studies employing site-directed mutagenesis or tandem mass spectrometry analysis would be valuable for definitive identification of the grafting sites and further optimization of the conjugation strategy.

### 4.2. Synergistic Antibacterial Mechanism

The antibacterial evaluation revealed a clear synergistic effect between chitosan and nisin, providing both quantitative evidence and mechanistic insights. Grafted products achieved MIC values significantly lower than Na-Cs alone (132.4 vs. 486.5 μg/mL against *E. coli*, 97.4 vs. 294.6 μg/mL against *S. aureus*) while maintaining substantial activity compared to pure nisin. The fact that Ni2-Cs, with only 14.35% nisin content, approached the efficacy of pure nisin treatment demonstrated exceptional efficiency of the grafted configuration.

The nucleic acid and protein leakage experiments provided direct mechanistic evidence for the synergistic mode of action. The progressive increase in leakage from Na-Cs to Ni1-Cs to Ni2-Cs, with absorbance at 260 nm and 280 nm showing parallel trends, confirmed that both chitosan and nisin contributed to membrane damage through complementary mechanisms. Chitosan, as a cationic polysaccharide, electrostatically interacts with negatively charged components on bacterial cell surfaces (lipopolysaccharides in Gram-negative bacteria, teichoic acids in Gram-positive bacteria), causing membrane depolarization and surface structure disruption. This initial damage creates favorable conditions for nisin action: the disrupted membrane becomes more permeable, facilitating nisin insertion, and the destabilized lipid bilayer offers reduced resistance to pore formation.

The spatial organization created by covalent grafting may amplify this synergistic effect. When nisin molecules are anchored onto chitosan chains, multiple nisin units are brought into close proximity. Upon chitosan adsorption to the bacterial surface, these locally concentrated nisin molecules can simultaneously attack the membrane, creating larger or more numerous pores than would result from randomly distributed free nisin. This “clustered attack” mechanism may explain why grafted products induced significantly more leakage than Na-Cs alone, despite having lower total nisin content than pure nisin treatments [33].

Particularly noteworthy was the improved efficacy against Gram-negative bacteria. The MIC ratio between *E. coli* and *S. aureus* decreased from 3.57 for pure nisin to 1.36 for Ni2-Cs, indicating a narrowed performance gap between bacterial types. This improvement likely stems from chitosan’s ability to disrupt the outer membrane of Gram-negative bacteria—a unique barrier absent in Gram-positive bacteria that normally limits nisin penetration. By breaching this outer membrane first, chitosan enables nisin to access the cytoplasmic membrane more effectively, thus extending the antimicrobial spectrum. This finding has important implications for food preservation applications where both Gram-positive and Gram-negative spoilage organisms are common [34].

### 4.3. Multifunctional Preservation Efficacy

The sea bass preservation trials validated the practical application potential by demonstrating comprehensive protection through multiple synergistic mechanisms. At day 15 of refrigerated storage, Ni2-Cs coatings maintained all critical quality indicators within acceptable ranges: total viable count at 5 log CFU/mL (below the 6 log CFU/mL safety limit), TVB-N at 30 mg/100 g (at the freshness threshold), springiness above 0.6, hardness at 13 N, and TBARS at 2.2 mg MDA/kg. In contrast, control samples had completely deteriorated by day 10, with TVC exceeding 7 log CFU/mL and TVB-N surpassing 55 mg/100 g.

This multifunctional preservation efficacy arose from the integration of antimicrobial, barrier, and antioxidant properties. The potent antibacterial activity, confirmed by in vitro tests, translated effectively to real food systems, suppressing microbial growth throughout storage. The sustained antibacterial effect, rather than an initial burst followed by rapid decline, suggested a controlled release mechanism: as the coating gradually hydrates and partially degrades in the aqueous fish surface environment, nisin is progressively released, maintaining inhibitory concentrations over extended periods. This sustained release is advantageous over direct nisin application, where rapid diffusion and degradation often limit efficacy duration [31].

The rheological properties of the coating solutions contributed significantly to their practical applicability. The shear-thinning behavior exhibited by all chitosan-based samples is particularly advantageous for food coating applications. During the coating process, which involves high shear conditions such as dipping or mechanical spreading, the reduced apparent viscosity facilitates uniform spreading and effective adhesion to the irregular surface of sea bass filets. After application, under low shear conditions, the viscosity recovers, enabling the coating to remain stable on the product surface and form a continuous protective film without dripping. Notably, the grafted products (Ni1-Cs and Ni2-Cs) exhibited lower viscosity than native chitosan while retaining shear-thinning characteristics, suggesting improved processability and ease of coating application without compromising film-forming ability.

Beyond antimicrobial action and favorable rheological behavior, the physical barrier properties of the coating played crucial roles in quality maintenance. The improved water vapor barrier (WVP reduced from 8.65 to 7.28 g·mm/m^2^·d·kPa) helped retain moisture in fish tissues, which directly impacts texture preservation. The maintenance of springiness (>0.6) and hardness (>13 N) at day 15, compared to dramatic declines in uncoated samples (0.4 and 6 N, respectively), demonstrated the effectiveness of moisture retention. Additionally, the oxygen barrier properties retarded lipid oxidation, as evidenced by TBARS values. Fish muscle is rich in polyunsaturated fatty acids highly susceptible to oxidative rancidity, which not only produces off-flavors, but also degrades nutritional value and produces toxic compounds. The coating’s oxygen barrier, combined with potential antioxidant activity of nisin peptide (certain amino acid residues like His and Cys can scavenge free radicals), provided dual protection against lipid deterioration.

The controlled pH increase (from 6.6 to 7.4 in Ni2-Cs vs. 8.0 in control) and reduced TVB-N accumulation further illustrated the comprehensive preservation mechanism. Both pH rise and TVB-N formation result primarily from microbial metabolism (deamination and decarboxylation of amino acids and nucleotides), with secondary contributions from endogenous enzyme activity. The coating’s suppression of microbial growth reduced exogenous protease production, while the physical barrier may also restrict endogenous enzyme activity by limiting substrate availability or oxygen required for certain degradative pathways. The fact that TVB-N in Ni2-Cs remained at the freshness limit (30 mg/100 g) while control exceeded it nearly two-fold (55 mg/100 g) indicated effective control of protein degradation from both microbial and enzymatic sources [1].

The superior performance of grafted coatings over pure nisin or Na-Cs alone confirmed that covalent integration achieved synergistic preservation effects exceeding simple additive combinations. Pure nisin, despite its potent antimicrobial activity, lacks film-forming ability and provides no physical barrier; Na-Cs forms films, but has limited antibacterial efficacy. Only through covalent grafting were these complementary functionalities unified into a single material, enabling “1 + 1 > 2” synergistic performance in real food preservation applications.

### 4.4. Implications and Future Directions

This study demonstrates that enzymatic modification under ultra-low-temperature conditions provides a green and effective strategy for developing multifunctional food preservation materials. The success of nisin-grafted chitosan suggests broader applicability of this approach for grafting other bioactive peptides (such as ε-polylysine, lactoferricin, or plant-derived antimicrobial peptides) onto polysaccharide carriers, potentially creating a library of tailored antibacterial materials for diverse food preservation needs.

However, several aspects require further investigation before commercial implementation. First, the economic feasibility must be assessed. Nisin, although FDA-approved and widely used, has relatively high costs compared to conventional preservatives. Optimization of grafting efficiency to reduce nisin usage while maintaining efficacy, or exploration of less expensive antimicrobial peptide alternatives, would improve commercial viability. Second, the long-term stability of grafted products under various storage conditions (light exposure, temperature fluctuations, humidity variations) needs systematic evaluation to ensure consistent performance throughout the product shelf life. Third, while chitosan and nisin are both GRAS (Generally Recognized As Safe) substances, the potential effects of covalent grafting on their metabolism and degradation pathways warrant investigation through in vivo studies to confirm food safety. Fourth, sensory attributes deserve attention: the introduction of nisin and reduced film transparency may affect product appearance and consumer acceptance, requiring optimization through formulation adjustments or incorporation of plasticizers and opacifiers.

From a mechanistic perspective, the precise molecular interactions underlying the synergistic effects warrant deeper investigation. Advanced techniques such as molecular dynamics simulations could elucidate the conformational changes in nisin upon grafting and predict optimal grafting sites and densities. High-resolution microscopy (atomic force microscopy, transmission electron microscopy) could visualize the membrane-disrupting processes in real-time, providing direct evidence for the proposed “disruption-perforation” mechanism. Understanding these molecular details would enable rational design of next-generation materials with optimized antimicrobial efficiency.

Finally, expansion of the application scope beyond sea bass to other perishable foods (crustaceans, fruits, vegetables, fresh-cut produce) would demonstrate the versatility of this technology and facilitate its adoption across the food industry. Each food matrix presents unique challenges in terms of indigenous microflora, biochemical composition, and surface characteristics, requiring tailored coating formulations. Systematic investigation of these applications would establish comprehensive guidelines for implementing nisin-grafted chitosan coatings in diverse food preservation scenarios.

## 5. Conclusions

This study proposed and validated four scientific hypotheses regarding papain-catalyzed nisin–chitosan conjugation under ultra-low-temperature conditions.

First, the hypothesis that papain can catalyze amidation between nisin’s carboxyl groups and chitosan’s amino groups was confirmed by FTIR, XRD, and XPS analyses, which provided unambiguous evidence for covalent amide bond formation. Second, the hypothesis that nisin grafting would disrupt chitosan’s crystalline structure and improve its processability was validated, as grafted products exhibited dramatically enhanced solubility and film-forming properties. Third, the proposed synergistic antibacterial mechanism combining chitosan’s membrane-disrupting capability with nisin’s pore-forming action was supported by the nucleic acid and protein leakage assays, demonstrating a “disruption-perforation” mechanism that enhanced antibacterial efficiency. Fourth, the practical application hypothesis was confirmed through sea bass preservation trials, where nisin-grafted chitosan coatings effectively extended shelf life to 15 days by maintaining quality indicators within acceptable limits.

These findings establish ultra-low-temperature enzymatic grafting as a green and effective strategy for developing multifunctional food preservation materials, with the temperature-dependent catalytic behavior of papain being essential for achieving amide bond formation while preserving nisin’s bioactive structure. Future research should focus on optimizing grafting parameters, exploring applications in other perishable food systems, and conducting comprehensive safety evaluations for commercial implementation.

## Figures and Tables

**Figure 1 foods-14-04227-f001:**
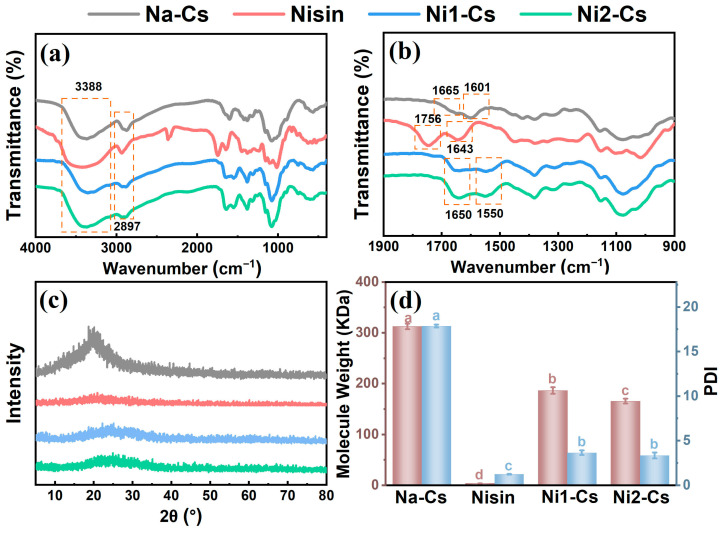
FTIR, XRD patterns, and molecular weight of samples. (**a**) and (**b**): FTIR spectra; (**c**): XRD spectra; (**d**): weight–average molecular weight of samples. Different letters indicate statistically significant differences between groups at *p* ≤ 0.05.

**Figure 2 foods-14-04227-f002:**
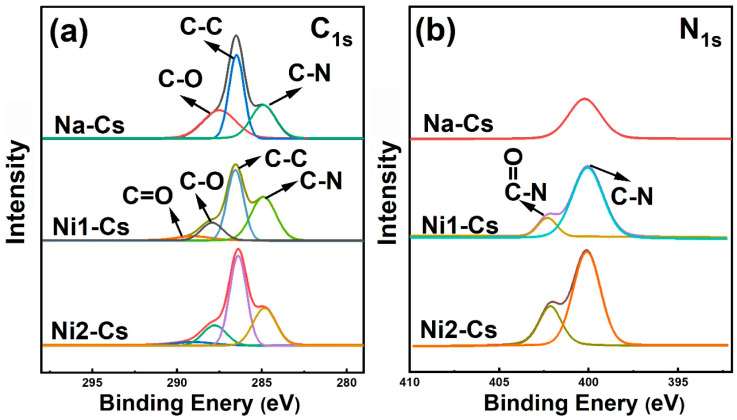
XPS analysis of samples. (**a**): C_1s_ high-resolution spectrum; (**b**): N1s high-resolution spectrum.

**Figure 3 foods-14-04227-f003:**
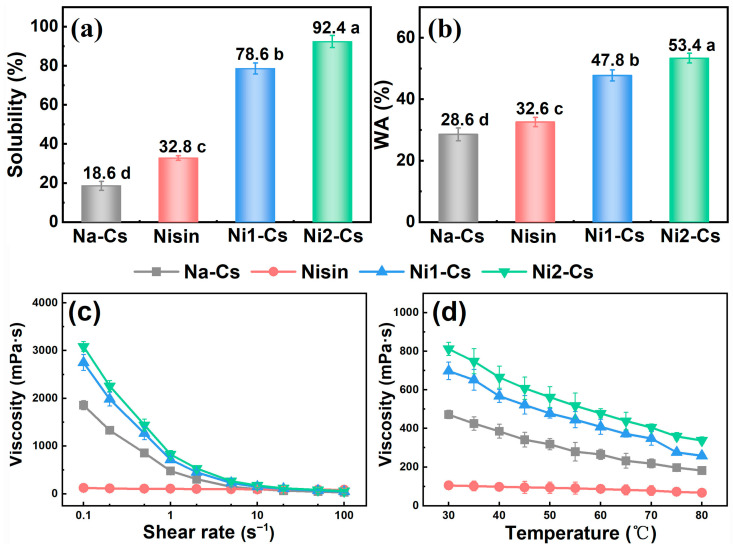
Solubility, water absorption, and viscosity behavior of samples. (**a**): solubility; (**b**): water absorption; (**c**): shear rate–viscosity behavior; (**d**): temperature–viscosity behavior. Different letters indicate statistically significant differences between groups at *p* ≤ 0.05.

**Figure 4 foods-14-04227-f004:**
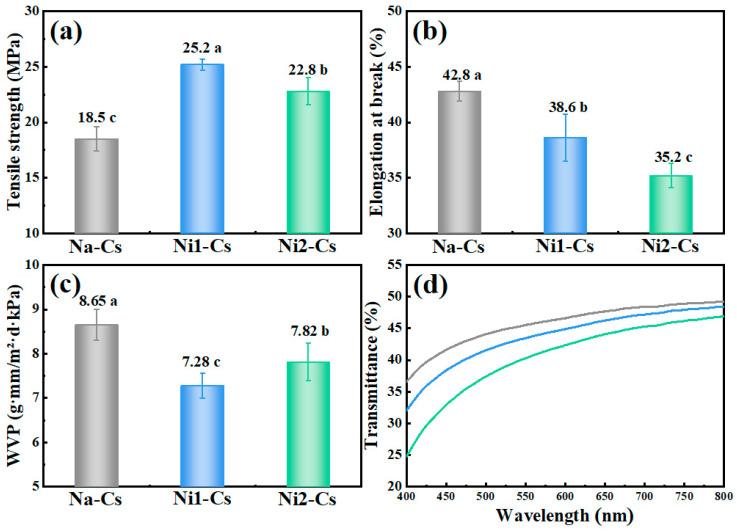
Tensile strength (**a**), elongation at break (**b**), water vapor permeability (**c**), and light transmittance (**d**) of samples. Different letters indicate statistically significant differences between groups at *p* ≤ 0.05.

**Figure 5 foods-14-04227-f005:**
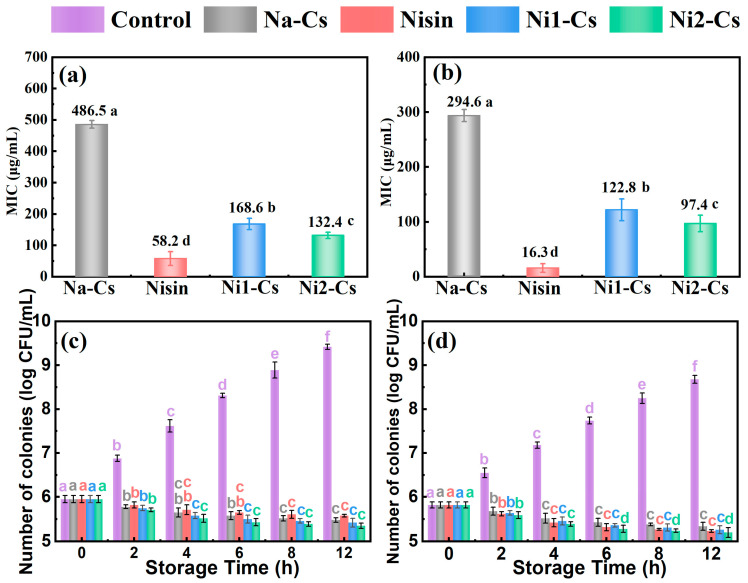
Minimum inhibitory concentration and antibacterial kinetics of samples. (**a**): minimum inhibitory concentration against *E. coli*; (**b**): minimum inhibitory concentration against *S. aureus*; (**c**): antibacterial kinetics against *E. coli*; (**d**): antibacterial kinetics against *S. aureus.* Different letters indicate statistically significant differences (*p* ≤ 0.05).

**Figure 6 foods-14-04227-f006:**
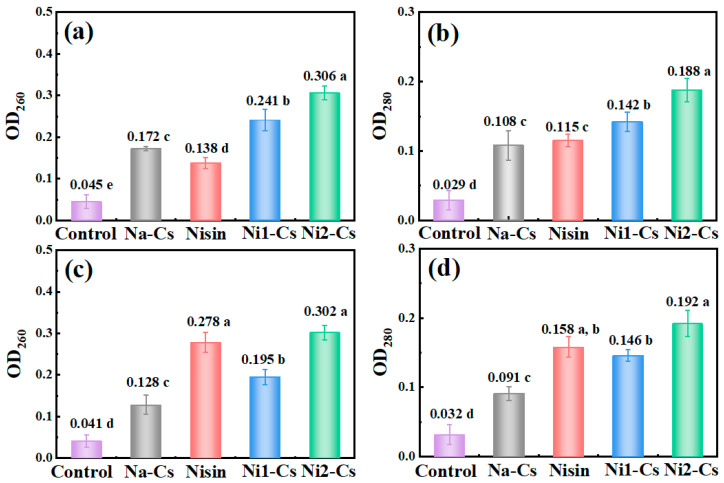
Nucleic acid and protein leakage during bacterial culture. (**a**): absorbance changes at 260 nm during *E. coli* culture; (**b**): absorbance changes at 280 nm during *E. coli* culture; (**c**): absorbance changes at 260 nm during *S. aureus* culture; (**d**): absorbance changes at 260 nm during *S. aureus* culture. Different letters indicate statistically significant differences between groups at *p* ≤ 0.05.

**Figure 7 foods-14-04227-f007:**
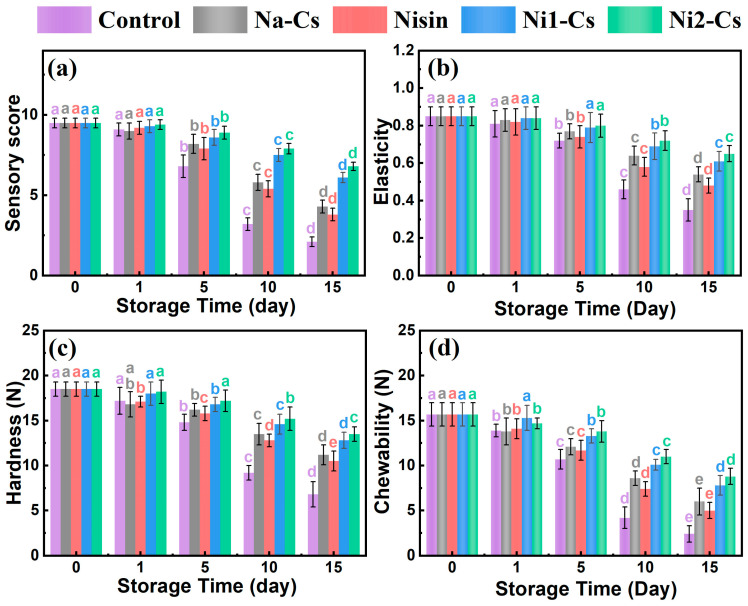
Sensory evaluation and texture analysis during sea bass storage. (**a**): sensory score; (**b**): springiness; (**c**): hardness; (**d**): chewiness. Different letters indicate statistically significant differences between groups at *p* ≤ 0.05.

**Figure 8 foods-14-04227-f008:**
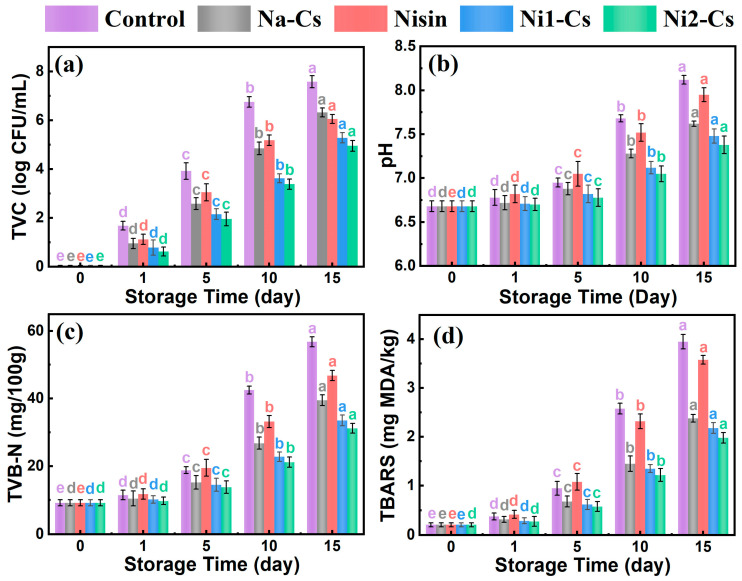
Microbiological and chemical component analysis during sea bass storage (**a**): total viable count; (**b**): pH; (**c**): TVB-N; (**d**): TBARS. Different letters indicate statistically significant differences between groups at *p* ≤ 0.05.

## Data Availability

The original contributions presented in the study are included in the article; further inquiries can be directed to the corresponding authors.

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
