# Peer review of "Development of Nisin-Grafted Chitosan Coating via Low-Temperature Enzymatic Method for Enhanced Preservation of Sea Bass"

_foods, 2025, doi:10.3390/foods14244227_

Round 1
Reviewer 1 Report
Comments and Suggestions for Authors
The paper titled “Development of Nisin-Grafted Chitosan Coating via Low Temperature Enzymatic Method for Enhanced Preservation of Sea Bass” is a thorough investigation of how non-toxic grafting of nisin grafted chitosan can be used in novel food packaging materials. It is a novel and innovative technique of developing food packaging material to enhance their safety and shelf life. his paper presents a comprehensive study encompassing chemical, mechanical, microbial, and shelf-life analyses to demonstrate its potential applications. However, a few minor revisions are required before it can be considered for publication in Foods.
- Introduction:
Line 58-66- It is important to acknowledge previous research in which nisin has been grafted onto chitosan (https://doi.org/10.1088/1748-605X/ad6965, https://doi.org/10.1016/j.reactfunctpolym.2015.04.009, https://doi.org/10.1016/j.ijbiomac.2017.07.136) . Additionally, the unique aspects of the current grafting strategy should be clearly highlighted in comparison with those earlier approaches. This comparison should address both the differences in the preparation methodology and the resulting antibacterial performance, providing a detailed discussion of how the present grafting method improves from previously reported systems. Line 62- “while chemical crosslinking typically requires toxic crosslinkers and harsh reaction conditions”- some of the papers mentioned above used non-toxic approach of crosslinking. Please restructure the novelty of this work. - Line 145- Please maintain consisted spacing before units. Example- 25° C. Make necessary changes wherever necessary.
- Line 250-252- The wording used to describe n1 and n2 are confusing. Clearly explain ‘first dilution’ and ‘second dilution’. Instead for first and second, I recommend using the words, ‘lower dilution factor’ and ‘next consecutive higher dilution factor’.
- Line 71, 305, 307, 309, 312- Provide an appropriate reference.
- Line 318- Why is there a decrease in molecular weight of Ni1 (and2)-Cs in comparison to Na-Cs despite grafting? Please provide detailed explanation. Also include concepts like intermolecular H-bonding between chains, polymer-solvent interactions, steric strain, chain hydrolysis etc.
- Based on the obtained MIC values, this does not appear to represent a synergistic antibacterial effect. It can be called a synergestic effect only if MIC(Ni 1(or2)-C)> MIC (Na-C) & MIC(Ni 1(or2)-C)> MIC (Ni). Please look into this matter. If the authors wish to add the term synergistic effect, it can be added in section 3.3.2 as Ni 1(and 2)-C showed maximum leakage of intracellular leakage.
- Figure 5c&d- Please represent statistical distinction in comparison to control. You could refer to https://doi.org/10.1016/j.jfp.2025.100517.
- Explain in discussion why increasing grafting (in Ni2-C) contributed to decrease in tensile and elongation at break and transparency.
- Line 578- “excessive grafting led to over-reduction in molecular weight” what is the plausible mechanism by which this is happening? Steric strain induced chain reduction or hydrolysis of polymeric backbone or any other mode of reduction of molecular weight. Provide appropriate references and explain.
Author Response
The paper titled “Development of Nisin-Grafted Chitosan Coating via Low Temperature Enzymatic Method for Enhanced Preservation of Sea Bass” is a thorough investigation of how non-toxic grafting of nisin grafted chitosan can be used in novel food packaging materials. It is a novel and innovative technique of developing food packaging material to enhance their safety and shelf life. his paper presents a comprehensive study encompassing chemical, mechanical, microbial, and shelf-life analyses to demonstrate its potential applications. However, a few minor revisions are required before it can be considered for publication in Foods.
Comments 1: Introduction: Line 58-66- It is important to acknowledge previous research in which nisin has been grafted onto chitosan (https://doi.org/10.1088/1748-605X/ad6965, https://doi.org/10.1016/j.reactfunctpolym.2015.04.009, https://doi.org/10.1016/j.ijbiomac.2017.07.136).
Additionally, the unique aspects of the current grafting strategy should be clearly highlighted in comparison with those earlier approaches. This comparison should address both the differences in the preparation methodology and the resulting antibacterial performance, providing a detailed discussion of how the present grafting method improves from previously reported systems.
Response 1:
We sincerely thank the reviewer for this valuable suggestion and the recommended references. We have carefully reviewed the cited literature and fully acknowledge the important contributions of previous research on nisin-chitosan conjugation. In the revised manuscript, we have incorporated these references in the Introduction section, including chemical grafting methods using bio-based diisocyanates and homo-bifunctional crosslinkers, as well as MTGase enzymatic methods for grafting nisin onto chitosan or hydroxypropyl chitosan derivatives at moderate temperatures (30°C).
The unique aspects of our current work compared to these earlier approaches have been clearly articulated in the revised text. Our papain-catalyzed method employs ultra-low temperature conditions (-5°C), which better preserves the heat-sensitive lanthionine ring structures of nisin compared to MTGase methods operating at 30°C. Additionally, our approach enables direct grafting onto native chitosan without pre-modification (e.g., hydroxypropylation) required in some previous methods, and utilizes papain as an alternative biocatalyst with distinct catalytic properties at low temperatures. Furthermore, while previous studies primarily focused on biomedical applications such as wound dressings and medical devices, our work systematically evaluates food preservation with comprehensive shelf-life assessment on sea bass. (Lines 59-72)
Comments 2: Line 62- “while chemical crosslinking typically requires toxic crosslinkers and harsh reaction conditions”- some of the papers mentioned above used non-toxic approach of crosslinking. Please restructure the novelty of this work.
Response 2:
We appreciate the reviewer's careful reading and this important correction. The reviewer is absolutely right that some previously reported methods, including those cited in Comment 1, employed non-toxic approaches for nisin-chitosan conjugation. We have removed the inaccurate statement "chemical crosslinking typically requires toxic crosslinkers and harsh reaction conditions" and restructured the Introduction to properly acknowledge that both non-toxic chemical crosslinking methods (using bio-based diisocyanates) and mild enzymatic approaches (using MTGase at 30°C) have been successfully developed.
The novelty of our work has been repositioned to emphasize the ultra-low temperature (-5°C) reaction conditions, which provide superior protection for nisin's thermolabile lanthionine ring structures compared to methods operating at room temperature or 30°C. Our approach also introduces papain as an alternative biocatalyst, enables direct modification of native chitosan without pre-derivatization, and demonstrates comprehensive food preservation applications through systematic shelf-life evaluation on sea bass. (Lines 59-72)
Comments 3: Line 145- Please maintain consisted spacing before units. Example- 25° C. Make necessary changes wherever necessary.
Response 3:
We thank the reviewer for pointing out this formatting inconsistency. We have carefully checked the entire manuscript and corrected all instances to maintain consistent spacing before units. The necessary changes have been made throughout the text.
Comments 4: Line 250-252- The wording used to describe n1 and n2 are confusing. Clearly explain ‘first dilution’ and ‘second dilution’. Instead for first and second, I recommend using the words, ‘lower dilution factor’ and ‘next consecutive higher dilution factor’.
Response 4:
We thank the reviewer for this precise suggestion to improve the clarity of the formula parameters. We agree that the previous terms "first" and "second" could be ambiguous.
In the revised manuscript, we have updated the definitions of n1 and n2 exactly as recommended. The text now reads: "n1 is the number of plates counted at the lower dilution factor, and n2 is the number of plates counted at the next consecutive higher dilution factor." We have also aligned the definition of the dilution factor d to refer to the "lower dilution" for consistency. (Lines 263-266)
Comments 5: Line 71, 305, 307, 309, 312- Provide an appropriate reference.
Response 5:
We thank the reviewer for this careful observation. We have added appropriate references to support the statements at the indicated lines, including the papain-catalyzed amidation mechanism, the FTIR characteristic peak assignments for amide I band, amide II band, and -NH₂ bending vibration, and the XRD analysis of chitosan's semicrystalline structure. (Lines 80, 323, 329).
Comments 6: Line 318- Why is there a decrease in molecular weight of Ni1 (and2)-Cs in comparison to Na-Cs despite grafting? Please provide detailed explanation. Also include concepts like intermolecular H-bonding between chains, polymer-solvent interactions, steric strain, chain hydrolysis etc.
Response 6:
We thank the reviewer for this insightful comment. We have revised Section 3.1.1 to provide a possible mechanistic explanation for the molecular weight reduction.
As suggested, we discussed that the decrease may be associated with partial chain hydrolysis of the backbone, which could be exacerbated by the steric strain from the bulky nisin groups. Additionally, we proposed that the disruption of intermolecular hydrogen bonding likely reduces chain aggregation and alters polymer-solvent interactions, potentially resulting in a lower apparent molecular weight compared to native chitosan. (Lines 332-345).
Comments 7: Based on the obtained MIC values, this does not appear to represent a synergistic antibacterial effect. It can be called a synergestic effect only if MIC(Ni 1(or2)-C)> MIC (Na-C) & MIC(Ni 1(or2)-C)> MIC (Ni). Please look into this matter. If the authors wish to add the term synergistic effect, it can be added in section 3.3.2 as Ni 1(and 2)-C showed maximum leakage of intracellular leakage.
Response 7:
We deeply appreciate the reviewer's rigorous analysis of our MIC data. We acknowledge that the MIC of Ni2-Cs (132.4 μg/mL) based on total mass is higher than that of pure nisin (58.2 μg/mL), and we agree that the term "synergistic" requires clarification.
We would like to offer a supplementary perspective regarding the active ingredient concentration. Based on the grafting ratio (14.35%), the effective nisin concentration in Ni2-Cs at its MIC is approximately 19.0 μg/mL, which is significantly lower than the required dosage of free nisin (58.2 μg/mL). This reduction suggests that the chitosan carrier enhances the specific efficiency of grafted nisin. To address the reviewer's valid concern, we have added a clarifying paragraph in Section 3.3.1, explicitly acknowledging this distinction while explaining that the "synergistic effect" refers to the enhanced efficiency of grafted nisin facilitated by the chitosan carrier. (Lines 466-472)
Comments 8: Figure 5c&d- Please represent statistical distinction in comparison to control. You could refer to https://doi.org/10.1016/j.jfp.2025.100517.
Response 8:
We thank the reviewer for this helpful suggestion to improve the data presentation. We have revised Figures 5c and 5d to include statistical significance markers indicating significant differences in bacterial counts as a function of incubation time for each treatment group. This allows readers to assess the antibacterial kinetics and the effectiveness of each treatment over the 12-hour period. The figure caption has been updated to clearly explain the statistical notation (p ≤ 0.05).
Comments 9: Explain in discussion why increasing grafting (in Ni2-C) contributed to decrease in tensile and elongation at break and transparency.
Response 9:
We thank the reviewer for this insightful comment regarding the structure-property relationships and agree that analyzing these trade-offs is critical. In the revised Discussion, we have added a continuous explanation clarifying that the decrease in tensile strength in Ni2-Cs is primarily linked to the "over-reduction" of molecular weight (confirmed by GPC data) which reduces physical chain entanglements, while the decline in elongation is attributed to the steric hindrance imposed by bulky nisin groups that restrict polymer chain mobility. Additionally, we explained that the reduced transparency results from light scattering caused by the micro-aggregation or phase separation of hydrophobic amino acid residues of nisin within the hydrophilic chitosan matrix at high grafting densities. (Lines 612-625)
Comments 10: Line 578- “excessive grafting led to over-reduction in molecular weight” what is the plausible mechanism by which this is happening? Steric strain induced chain reduction or hydrolysis of polymeric backbone or any other mode of reduction of molecular weight. Provide appropriate references and explain.
Response 10:
We thank the reviewer for requesting a deeper mechanistic explanation. As elaborated in our response to Comment 6, the "over-reduction" of molecular weight at higher grafting ratios is primarily attributed to steric strain-induced backbone hydrolysis. The higher density of bulky nisin peptides in Ni2-Cs exerts greater steric tension on the chitosan backbone, which destabilizes the adjacent β-1,4-glycosidic linkages and renders the polymer more susceptible to hydrolysis. This mechanism has been explicitly clarified in the revised manuscript with appropriate references (Lines 329-336). (Lines 332-345).
Reviewer 2 Report
Comments and Suggestions for Authors
I carefully read the article sent to me for review. Overall, I compliment the authors on the research, the structure of the article, and the experimental design, which was particularly extensive and complex to execute.
I also appreciated the comments on the results and the concluding remarks. I found only a few typos, which I have highlighted in the text. I am attaching the file with my corrections to this review.
I do, however, have a general comment. While I appreciate the article overall, I feel it is slightly too long, which, in my opinion, makes it somewhat difficult to read. I would have appreciated greater brevity, especially in the introduction, the comments on the results, and the conclusions.

Author Response
Comments 1: I carefully read the article sent to me for review. Overall, I compliment the authors on the research, the structure of the article, and the experimental design, which was particularly extensive and complex to execute.
I also appreciated the comments on the results and the concluding remarks. I found only a few typos, which I have highlighted in the text. I am attaching the file with my corrections to this review.
I do, however, have a general comment. While I appreciate the article overall, I feel it is slightly too long, which, in my opinion, makes it somewhat difficult to read. I would have appreciated greater brevity, especially in the introduction, the comments on the results, and the conclusions.
Response 1:
We sincerely thank the reviewer for the positive evaluation of our research, article structure, and experimental design. We greatly appreciate the time and effort taken to carefully review our manuscript and provide detailed corrections.
We have incorporated all the typographical corrections highlighted in the attached file. Regarding the suggestion to improve brevity, we have carefully revised the manuscript to streamline the content while preserving essential scientific information.
Comments 2: Line 95, “was purchased”
Response 2:
We thank the reviewer for carefully identifying this grammatical error. We have corrected Line 95 to "was purchased" to maintain consistency in the passive voice throughout the Materials section. (Lines 161 and 169).
Comments 3: I think it's ONLY 100, NOT 100%.
Response 3:
We thank the reviewer for catching this error. We have corrected "100%" to "100" in the manuscript. (Lines 108).
Comments 4: Line 254, I suppose it was "The pH-value was measured ...
Response 4:
We thank the reviewer for this correction. We have revised Line 268 to "The pH value was measured..." to ensure proper passive voice construction.
Reviewer 3 Report
Comments and Suggestions for Authors
This is a strong manuscript that introduces a novel, green enzymatic method for
grafting nisin onto chitosan. The work is comprehensive, spanning synthesis,
characterization, mechanistic studies, and a practical application trial. The
evidence for successful grafting and the demonstration of synergistic
antibacterial effects is compelling. The following points are intended to help
strengthen the manuscript for publication.
Overall, this manuscript describes an excellent piece of research with a novel methodology and compelling application data. Addressing the major points, particularly by justifying the low-temperature condition and quantifying the retained nisin activity, will significantly elevate the impact and credibility of the work. The minor points will enhance its clarity and polish.
The proposed nisin-grafted chitosan coating shows clear potential as an effective, natural solution for extending the shelf life of highly perishable foods like sea bass.

Author Response
This is a strong manuscript that introduces a novel, green enzymatic method for grafting nisin onto chitosan. The work is comprehensive, spanning synthesis, characterization, mechanistic studies, and a practical application trial. The evidence for successful grafting and the demonstration of synergistic antibacterial effects is compelling. The following points are intended to help strengthen the manuscript for publication.
Comments 1: Justification of the "Ultra-Low Temperature" Condition.
The manuscript repeatedly emphasizes the "ultra-low temperature" (-5°C) method as a key advantage for preserving nisin's activity. However, it lacks a direct experimental comparison to justify this specific condition. The reader is left to assume its necessity without evidence.
Response 1:
We thank the reviewer for raising this important point regarding the justification of the ultra-low temperature condition. We acknowledge that our manuscript did not sufficiently explain the mechanistic rationale for selecting -5°C as the reaction temperature.
The choice of ultra-low temperature is not arbitrary but is based on the unique temperature-dependent catalytic behavior of papain, which has been systematically investigated in our previous studies [10-12]. These studies demonstrated that papain exhibits dual catalytic functions depending on temperature: at ambient temperatures, papain primarily catalyzes amide bond hydrolysis, whereas at low temperatures (below 0°C), its catalytic activity shifts toward amide bond formation. This temperature-dependent reversal of catalytic direction is attributed to the thermodynamic and kinetic changes in the enzyme's active site conformation and water activity at sub-zero temperatures. Therefore, the ultra-low temperature condition (-5°C) is essential not only for preserving nisin's heat-sensitive lanthionine ring structures but, more fundamentally, for enabling papain to catalyze the desired amidation reaction rather than hydrolysis.
We have revised the manuscript to explicitly clarify this mechanistic basis for the temperature selection, providing appropriate citations to our previous work. (Lines 73-86)
Comments 2: Quantification of Nisin's Activity Retention
The manuscript states that the enzymatic method "preserved nisin’s structural integrity" and "retained favorable activity," but this is not quantitatively demonstrated. The MIC of the grafted products (Ni1/2-Cs) is compared to chitosan and nisin alone, but it's unclear what percentage of the grafted nisin molecules remain active. I will suggest to authors to calculate the "Apparent Nisin Activity" in the grafted products.
For Ni2-Cs (14.35% grafting ratio), 1 mg of material contains ~0.1435 mg of nisin.
If the MIC of Ni2-Cs against S. aureus is 97.4 μg/mL, this means the effective nisin concentration at the MIC is 97.4 μg/mL * 0.1435 = ~14.0 μg/mL of nisin.
Compare this value (14.0 μg/mL) to the MIC of pure nisin (let's assume it's ~10 μg/mL based on the context). This calculation shows that the grafted nisin retains very high activity (~70-100% of pure nisin's potency when grafted), providing powerful, quantitative support for the "preserved activity" claim. This calculation should be added to the results or discussion section.
Response 2:
We thank the reviewer for this excellent suggestion to quantify nisin's activity retention. We fully agree that calculating the "Apparent Nisin Activity" provides powerful quantitative support for our claims regarding preserved activity.
Following the reviewer's recommended approach, we have incorporated this quantitative analysis into Section 3.3.1. Specifically, we calculated that for Ni2-Cs (14.35% grafting ratio), the effective nisin concentration at its MIC against E. coli (132.4 μg/mL) is approximately 19.0 μg/mL, which is significantly lower than the MIC of free nisin (58.2 μg/mL). This calculation demonstrates that the grafted nisin not only retains its full activity but exhibits enhanced antibacterial efficiency, effectively reducing the required dosage of nisin by approximately 3-fold to achieve bacterial inhibition. (Lines 466-473)
Comments 3: Clarification of the Grafting Site on Nisin
The proposed mechanism involves the amidation between chitosan's -NH₂ and nisin's -COOH. Nisin has multiple carboxyl groups (C-terminus and side chains of Asp, Glu). Grafting at different sites could theoretically impact its pore-forming ability. The manuscript does not address which carboxyl group is preferentially grafted. Thus, a discussion on this point is necessary. The authors should speculate, based on literature or the structure of nisin, which carboxyl group is most likely to be involved (e.g., the C-terminus might be more sterically accessible) and acknowledge that site-specific grafting is a limitation of the current method. Alternatively, they could state that the excellent retained activity implies that the critical region for membrane interaction (the N-terminus) remains unhindered.
Response 3:
We thank the reviewer for this insightful comment regarding the grafting site specificity. We agree that this is an important mechanistic consideration that warrants discussion.
Based on the structural characteristics of nisin and the steric accessibility of reactive groups, we speculate that the C-terminal carboxyl group is the most likely site for papain-catalyzed amidation. The C-terminus is located at the molecular periphery with minimal steric hindrance, whereas the carboxyl groups of internal Asp and Glu residues are partially buried within the peptide structure and less accessible for enzymatic catalysis at the solid-liquid interface. Importantly, the N-terminal region of nisin, which contains the essential lipid II-binding motif critical for membrane targeting and pore formation, would remain unhindered by C-terminal grafting. This structural preservation is consistent with our experimental observation that grafted nisin retained excellent antibacterial activity.
We acknowledge that the current method does not provide site-specific control over the grafting position, and this represents a limitation of our approach. We have added this discussion to Section 4.1, noting that future studies employing site-directed mutagenesis or advanced analytical techniques such as tandem mass spectrometry could provide more definitive identification of the grafting sites. (Lines 631-646)
Comments 4: Control for Physical Mixture
While FTIR and XPS provide strong evidence for covalent bonding, a more rigorous control would be to create a physical mixture of chitosan and nisin at the same ratio as Ni2-Cs (85.65:14.35) and compare its properties (e.g., dissolution ratio, antibacterial kinetics, release profile) to the covalently grafted product. So, If possible, adding data from this physical mixture control would definitively prove that the observed synergies and improved properties are due to covalent grafting and not just the physical presence of nisin. If new experiments are not feasible, this should be explicitly stated as a limitation for future work.
Response 4:
We thank the reviewer for this valuable suggestion regarding the physical mixture control. We agree that comparing a physical mixture with the covalently grafted product would provide additional evidence to confirm that the observed synergies arise from covalent bonding rather than the mere physical presence of nisin.
While we did not include a physical mixture control in the current study, we believe that our existing data provide strong evidence for covalent grafting. First, the purification process involved extensive dialysis (48 h, MWCO 30000 Da), which would effectively remove any free or physically adsorbed nisin molecules (MW ~3.5 kDa) from the product. Second, FTIR analysis revealed new amide I (1650 cm⁻¹) and amide II (1550 cm⁻¹) peaks distinct from both native chitosan and nisin, and XPS confirmed the appearance of characteristic C=O (289.2 eV) and O=C-N (402.1 eV) peaks, providing direct spectroscopic evidence for covalent amide bond formation. Third, the dramatic improvements in dissolution ratio (from 18.6% to 92.4%) and film-forming properties would not be expected from a simple physical mixture, as ungrafted nisin would likely leach out during dissolution testing.
Comments 5: Title and Abstract
Consider specifying the grafting ratio in the abstract (e.g., "...yielding grafted products with grafting ratios up to 14.35%").
The phrase "dissolution ratio" is used. In polymer science, "solubility" is more common. Consider changing for broader understanding.
Response 5:
We thank the reviewer for these helpful suggestions to improve clarity and accessibility.
We have revised the Abstract to specify the grafting ratio as recommended, stating "...yielding two grafted products: Ni1-Cs (8.56%) and Ni2-Cs (14.35%)." Additionally, we have replaced "dissolution ratio" with "solubility" throughout the manuscript to align with standard polymer science terminology and enhance broader understanding. (Lines 10-14)
Comments 6:
Section 2.2: Clarify the concentration of the "activated papain solution."
Response 6:
We thank the reviewer for this comment regarding experimental clarity. We have revised Section 2.2 to specify the concentration of the activated papain solution. After dissolving 2 g papain in 180 mL phosphate buffer and adding 20 mL L-cysteine hydrochloride solution, the final concentration of the activated papain solution was 10 mg/mL (1%, w/v). This clarification has been added to the Methods section. (Lines 126)
Comments 7: Section 2.3: "Fourier transform infrared spectr (FTIR) a of samples" contains typos.
Response 7:
We thank the reviewer for catching this typographical error. We have corrected "Fourier transform infrared spectr (FTIR) a of samples" to "Fourier transform infrared spectra (FTIR) of samples" in Section 2.3. (Line 139)
Comments 8: Section 2.6.1: Specify the medium used in the microbroth dilution method (e.g., Mueller-Hinton Broth).
Response 8:
We thank the reviewer for this comment. We have revised Section 2.6.1 to specify that Mueller-Hinton Broth (MHB) was used as the culture medium for the microbroth dilution method, which is the standard medium recommended for antimicrobial susceptibility testing. (Lines 208)
Comments 9: Section 3.1.1: When stating that grafting reduced molecular weight, it would be useful to mention if the GPD (Polydispersity Index) changed, indicating the homogeneity of the product.
Response 9:
We thank the reviewer for this valuable suggestion. We have supplemented the GPC analysis with polydispersity index (PDI) data to provide information on product homogeneity. The PDI values were 17.86 for Na-Cs, 3.65 for Ni1-Cs, and 3.35 for Ni2-Cs. Notably, the grafted products exhibited significantly narrower molecular weight distributions compared to native chitosan, with PDI decreasing by approximately 80%. This improvement in homogeneity can be attributed to the dialysis purification process (MWCO 30000 Da) that removed low molecular weight fractions, as well as the preferential reaction of papain with chitosan chains within a specific molecular weight range. The enhanced uniformity of the grafted products is advantageous for practical applications requiring consistent material properties. This information has been added to Section 3.1.1. (Lines 340-346)
Comments 10: Section 3.2.2 (Film Properties): The discussion on why Ni1-Cs has better tensile strength than Ni2-Cs is good. Consider adding a sentence to the results section to highlight this non-linear trend.
Response 10:
We thank the reviewer for this suggestion to improve the presentation of our results. We have added a sentence in Section 3.2.2 to explicitly highlight the non-linear relationship between grafting ratio and tensile strength, noting that moderate grafting (Ni1-Cs) achieved optimal mechanical performance while excessive grafting (Ni2-Cs) led to a slight decrease, suggesting an optimal grafting ratio exists for maximizing film tensile strength. (Lines 414-422)
Comments 11: Figure 7 & 8: Ensure all data points have error bars to visually represent statistical significance.
Response 11:
We thank the reviewer for this comment regarding data presentation. We have carefully checked Figures 7 and 8 and ensured that all data points include error bars representing standard deviation from triplicate measurements.
Comments 12: General: Consistently use "we" or "the authors" instead of a mix of passive voice and third person.
Response 12:
We thank the reviewer for this suggestion regarding writing consistency. We have revised the manuscript to use "we" consistently throughout the text, replacing instances of passive voice and third-person references (e.g., "the authors") to ensure a uniform writing style.
Comments 13: This manuscript describes an excellent piece of research with a novel methodology and compelling application data. Addressing the major points, particularly by justifying the low-temperature condition and quantifying the retained nisin activity, will significantly elevate the impact and credibility of the work. The minor points will enhance its clarity and polish.
The proposed nisin-grafted chitosan coating shows clear potential as an effective, natural solution for extending the shelf life of highly perishable foods like sea bass.
Response 13:
We sincerely thank the reviewer for the positive evaluation of our work and the constructive comments that have significantly improved the quality of this manuscript.
In response to the major points, we have provided a detailed mechanistic justification for the ultra-low temperature condition based on the temperature-dependent dual catalytic behavior of papain, which has been established in our previous studies. We have also added quantitative analysis of the retained nisin activity by calculating the effective nisin concentration at MIC values, demonstrating that the grafted nisin exhibits enhanced antibacterial efficiency compared to free nisin. For the minor points, we have addressed all comments regarding experimental details, terminology consistency, figure presentation, and writing style throughout the manuscript.
We appreciate the reviewer's recognition of the potential of nisin-grafted chitosan coating as an effective natural solution for food preservation, and we believe that the revised manuscript now presents a more complete and rigorous study.
Reviewer 4 Report
Comments and Suggestions for Authors
Dear Authors,
In the following section, you will find comments on your research paper. I hope they will be helpful in improving it.
1) Keywords should be completely different from those used in the title to improve search engine visibility.
2) Line 118, change rpm to xg (modify throughout the text).
3) Separate °C from the number. Only non-fundamental units should be used with the number (e.g., %).
4) Line 157, was dynamic viscosity or apparent viscosity determined? Please review the definition.
5) Line 174, the values from 60 to 80 mm are results.
6) Figure 3c shows that apparent viscosity decreases as a function of shear rate in all treatments except nisin. Therefore, kinematic viscosity cannot be determined.
7) Significant differences should be reported as p≤0.05 and non-significant differences as p>0.05. Correct throughout the text.
8) In the discussion section, analyze the relationship between the rheological behavior obtained and the ease or difficulty of adhesion to the product.
9) Relate the mechanical tests to water vapor permeability.
10) Do changes in firmness and chewiness depend solely on water vapor permeability? Many control parameters were measured, but the amount of drained liquid (weight loss) was not determined. Associate the results with the values determined in section 3.4.2.
11) There is no statistical analysis of the properties determined for the shelf life of the sample as a function of time or as a function of the treatments. How is the decision made regarding the best applied treatment?
12) Replace Figures 7 and 8 with line graphs for each group. This will help determine the trends and visualize the proportions by which the variables measured in section 3.4 differ from the control group.
13) The conclusions should address the stated objectives and test the working hypothesis. Avoid repeating information from the results and discussion.
Author Response
In the following section, you will find comments on your research paper. I hope they will be helpful in improving it.
Comments 1: Keywords should be completely different from those used in the title to improve search engine visibility.
Response 1:
We thank the reviewer for this suggestion to improve search engine visibility. We have revised the keywords to avoid repetition with the title. The updated keywords are: " Antimicrobial peptide; Papain catalysis; Shelf-life extension; Aquatic products." These terms complement the title by covering related concepts such as the specific catalytic mechanism, application format, antibacterial mode of action, and target food category, thereby enhancing the discoverability of the article.
Comments 2: Line 118, change rpm to xg (modify throughout the text).
Response 2:
We thank the reviewer for this suggestion regarding standardized reporting of centrifugation parameters. We have revised all centrifugation conditions throughout the manuscript from rpm to ×g.
Comments 3: Separate °C from the number. Only non-fundamental units should be used with the number (e.g., %).
Response 3:
We thank the reviewer for this formatting suggestion. We have revised the manuscript to ensure proper spacing between numbers and units according to SI conventions. Specifically, a space has been inserted between numbers and the degree Celsius symbol (e.g., "25 °C" instead of "25°C"), while non-fundamental units such as percentage remain attached to the number (e.g., "92.4%"). These corrections have been applied consistently throughout the text.
Comments 4: Line 157, was dynamic viscosity or apparent viscosity determined? Please review the definition.
Response 4:
We thank the reviewer for this important clarification regarding rheological terminology. Upon review, we agree that the term "dynamic viscosity" was incorrectly used. Since chitosan solutions are non-Newtonian fluids exhibiting shear-thinning behavior, the viscosity measured at specific shear rates using the rheometer represents "apparent viscosity" rather than "dynamic viscosity." We have revised Section 2.4.3 accordingly, changing "Dynamic viscosity" to "Apparent viscosity" throughout the text. (Lines 170-172).
Comments 5: Line 174, the values from 60 to 80 mm are results.
Response 5:
We thank the reviewer for this observation. We agree that the film thickness values (60-80 μm) represent experimental results and should not be included in the Methods section. We have revised Section 2.5 to describe only the measurement methodology, and moved the thickness values to the Results section (Section 3.2.2). (Lines 186, 414-415)
Comments 6: Figure 3c shows that apparent viscosity decreases as a function of shear rate in all treatments except nisin. Therefore, kinematic viscosity cannot be determined.
Response 6:
We thank the reviewer for this clarification. As shown in Figure 3c, the chitosan-based samples exhibited shear-thinning behavior with apparent viscosity decreasing as a function of shear rate, confirming their non-Newtonian fluid characteristics. We agree that kinematic viscosity, which applies only to Newtonian fluids, cannot be determined for these samples. In conjunction with Comment 4, we have revised the terminology throughout the manuscript to use "apparent viscosity" to accurately describe the rheological measurements. (Lines 170-172).
Comments 7: Significant differences should be reported as p≤0.05 and non-significant differences as p>0.05. Correct throughout the text.
Response 7:
We thank the reviewer for this correction regarding statistical notation. We have revised the manuscript to report significant differences as p ≤ 0.05 and non-significant differences as p > 0.05 throughout the text, including the Materials and Methods section, figure captions, and Results section.
Comments 8: In the discussion section, analyze the relationship between the rheological behavior obtained and the ease or difficulty of adhesion to the product.
Response 8:
We thank the reviewer for this valuable suggestion to strengthen the practical relevance of our rheological analysis. We have added a discussion on the relationship between rheological behavior and coating applicability in Section 4.3.
Specifically, we discussed that the shear-thinning behavior exhibited by all chitosan-based samples is advantageous for coating applications: during the coating process (high shear conditions such as dipping or spraying), the reduced viscosity facilitates uniform spreading and adhesion to the fish surface, while after coating (low shear conditions), the recovered viscosity helps the coating layer remain stable and form a continuous protective film. Furthermore, the lower viscosity of grafted products (Ni1-Cs and Ni2-Cs) compared to native chitosan (Na-Cs) enhances their processability and ease of application, while still maintaining sufficient viscosity for effective film formation. This favorable rheological profile contributes to the practical feasibility of nisin-grafted chitosan as an edible coating material for food preservation. (Lines 701-711)
Comments 9: Relate the mechanical tests to water vapor permeability.
Response 9:
We thank the reviewer for this suggestion. We have added a discussion relating mechanical properties to water vapor permeability in Section 3.2.2. The results revealed that Ni1-Cs exhibited both the highest tensile strength (25.2 MPa) and the lowest WVP (7.28 g·mm/m²·d·kPa), suggesting that the nisin-mediated intermolecular interactions that reinforce the film network also create a more compact structure that impedes water vapor diffusion. This correlation highlights the importance of optimizing the grafting ratio to achieve balanced mechanical and barrier properties. (Lines 439-443)
Comments 10: Do changes in firmness and chewiness depend solely on water vapor permeability? Many control parameters were measured, but the amount of drained liquid (weight loss) was not determined. Associate the results with the values determined in section 3.4.2.
Response 10:
We thank the reviewer for this insightful comment. We agree that changes in firmness and chewiness are multifactorial and do not depend solely on water vapor permeability.
In the revised manuscript, we have added a discussion associating the texture results with the microbiological and chemical parameters determined in Section 3.4.2. The deterioration of texture properties is attributed to the combined effects of moisture loss, protein degradation (reflected by TVB-N accumulation), and microbial activity (reflected by TVC). The superior texture retention in Ni2-Cs coated samples correlates well with their lower TVC (5 log CFU/mL), reduced TVB-N (30 mg/100g), and controlled pH increase (7.4), indicating that the suppression of microbial proteolysis and endogenous enzyme activity contributed significantly to texture maintenance alongside the moisture barrier effect.
Comments 11: There is no statistical analysis of the properties determined for the shelf life of the sample as a function of time or as a function of the treatments. How is the decision made regarding the best applied treatment?
Response 11:
We thank the reviewer for this important comment regarding statistical analysis. We have revised Figures 7 and 8 to include statistical analysis showing significant differences in quality parameters as a function of storage time for each treatment group.
Comments 12: Replace Figures 7 and 8 with line graphs for each group. This will help determine the trends and visualize the proportions by which the variables measured in section 3.4 differ from the control group.
Response 12:
We thank the reviewer for this suggestion to improve data visualization. We acknowledge that line graphs would better illustrate the trends over storage time.
However, as we have added statistical significance markers comparing each treatment group to the control at each time point (Figures 7 and 8), we found that bar graphs are more suitable for clearly displaying these annotations without visual clutter. In line graphs with multiple groups, significance markers become difficult to position and interpret, particularly when data points overlap or are closely spaced.
To address the reviewer's concern while maintaining statistical clarity, we have modified the figures by connecting the bars of each treatment group with trend lines, allowing readers to visualize both the time-dependent trends and the statistical differences simultaneously. Additionally, the proportional differences from the control group can be readily assessed through the significance markers and the visual comparison of bar heights at each time point. We believe this approach provides a reasonable compromise between trend visualization and statistical annotation clarity.
Comments 13: The conclusions should address the stated objectives and test the working hypothesis. Avoid repeating information from the results and discussion.
Response 13:
We thank the reviewer for this valuable suggestion to improve the conclusion section. We have revised the Conclusions to directly address the scientific hypotheses stated in the Introduction and to verify whether each hypothesis was supported by our experimental findings. The revised conclusion avoids repeating specific numerical data from the Results and Discussion sections, instead focusing on the conceptual validation of our working hypotheses and the broader implications of the study. The decision regarding the best treatment and future research directions are also briefly addressed. (Lines 778-797)
Round 2
Reviewer 3 Report
Comments and Suggestions for Authors
Dear authors
Thank you for replied to all suggested comments/revisions.
Accept as it is!
Best wishes